# Early Classification of Time Series: A Survey and Benchmark

**Aurélien Renault**                                            *aurelien.renault@orange.com*
*Orange Research*
*AgroParisTech*

**Alexis Bondu**                                                *alexis.bondu@orange.com*
*Orange Research*

**Antoine Cornuéjols**                                          *antoine.cornuejols@agroparistech.fr*
*AgroParisTech*

**Vincent Lemaire**                                             *vincent.lemaire@orange.com*
*Orange Research*

**Reviewed on OpenReview:** *https: // openreview. net/ forum? id= bcNDYmBicK*

## Abstract

In many situations, the measurements of a studied phenomenon are provided sequentially, and the prediction of its class needs to be made as early as possible so as not to incur too high a time penalty, but not too early and risk paying the cost of misclassification. This problem has been particularly studied in the case of time series, and is known as Early Classification of Time Series (ECTS). Although it has been the subject of a growing body of literature, there is still a lack of a systematic, shared evaluation protocol to compare the relative merits of the various existing methods. In this paper, we highlight the two components of an ECTS system: *decision* and *prediction*, and focus on the approaches that separate them. This document begins by situating these methods within a principle-based taxonomy. It defines dimensions for organizing their evaluation and then reports the results of a very extensive set of experiments along these dimensions involving nine state-of-the-art ECTS algorithms. In addition, these and other experiments can be carried out using an open-source library in which most of the existing ECTS algorithms have been implemented (see https://github.com/ML-EDM/ml_edm).

## 1 Introduction

In hospital emergency rooms (Mathukia et al., 2015), in the control rooms of national or international power grids (Dachraoui et al., 2015), in government councils assessing critical situations, in many situations there is a *time pressure* to make early decisions. On the one hand, the longer a decision is delayed, the lower the risk of making the wrong decision, as knowledge of the problem increases with time. On the other hand, late decisions are generally more costly, if only because early decisions allow one to be better prepared.

A number of applications thus involve making decisions that optimize a trade-off between the accuracy of the prediction and its earliness. The problem is that favoring one usually works against the other. Greater accuracy comes at the price of waiting for more data. Such a compromise between the *Earliness* and the *Accuracy* of decisions has been particularly studied in the field of Early Classification of Time Series (ECTS), and introduced by Xing et al. (2008). ECTS consists in finding the *optimal time* to trigger the class prediction of an input time series observed over time.

As pointed out by Bondu et al. (2022), the ECTS problem is rooted in optimal stopping, of which it is a special case (Shepp, 1969; Ferguson, 1989), where the decision to be made concerns both: (*i*) *when* to stop receiving new measurements in order to (*ii*) *predict the class* of the incoming time series. To the best of our knowledge, Alonso González & Diez (2004) are the earliest explicitly mentioning *"classification when only part of the series are*

*presented to the classifier"*. Since then, several researchers have continued their efforts in this direction and have published a large number of research articles (Xing et al., 2009; Anderson et al., 2012; Dachraoui et al., 2015; Mori et al., 2017a; Schäfer & Leser, 2020; Lv & Hu, 2022; Ebihara et al., 2025).

However, despite the growing interest in ECTS over the last twenty years (Gupta et al., 2020; Akasiadis et al., 2024; Santos & Kern, 2016), there still remains a need for a shared taxonomy of approaches and an agreed well-grounded evaluation methodology (see Table 3 in Appendix A). **Guidelines** that we feel are important for a fair and informative comparison of existing ECTS approaches are listed below:

1. *Costs taken into account* for evaluating the performance of the proposed method *should be explicitly stated.* It seems natural to distinguish between the misclassification costs, and the delay cost, and to add them in order to define the cost of making a decision at time $t$. The delay cost may also depend on the true class $y$ and the predicted one $\hat{y}$, and a single cost function integrating misclassification and delay costs should then be used. For the sake of clarity, we keep the simple notation that distinguishes both cost functions in the rest of this paper.

2. The performance of the proposed methods *should be evaluated against a range of possible types of cost functions.* It is usual to evaluate "by default" the methods using a $\ell_{0-1}$ loss function that penalizes a wrong classification by a unity cost, and to consider a linear delay cost function. However, lots of applications rather involve unbalanced misclassification costs, and possibly also non-linear delay costs.

3. *The contributions of the various components of an ECTS algorithm should be as clearly delineated as possible.* The predominant approach to ECTS is to have a *decision* component which is in charge of evaluating the best moment to make the prediction about the class of the incoming time series, and a *classifier* one which makes the prediction itself. We call these methods "**separable methods**". In order to fairly compare the triggering methods at play, which are at the heart of ECTS, the classifier used should be the same for all methods in the experiments.
   Recent approaches mix the *decision* and the *prediction* components. This is the case of "end-to-end" methods using neural networks. These methods do not allow the evaluation of the merits of the decision component by itself, independently of the classification component, and they are accordingly not considered here.

4. *Performance obtained should be compared with that of "baseline" algorithms.* Failing to do this weakens any claim about the value of the proposed method. In the case of ECTS tasks, two naive baselines are: (1) make a prediction as soon as it is allowed, and (2) make a prediction after the entire time series has been observed. In our experiments reported in Section 4, we have added a third baseline less simple than the two aforementioned, which, to our knowledge has never been published as an original method. This is a confidence-based method where a decision is triggered as soon as the confidence for the likeliest class given $\mathbf{x}_t$ is greater than a threshold.

5. *The datasets used for training and testing should remove the biases* that are specific to the ECTS task and that may result in erroneous evaluations. A case in point, concerns the normalization often used in time series datasets. Dau et al. (2019) have reported that 71% of the reference time series classification datasets used to evaluate ECTS methods are made up of z-normalized time series, Not only, *this setting is not applicable in practice*, as z-normalization would require knowledge of the entire incoming time series, but it could also introduce a leakage of information from the future of the time series to the previous time steps which is detrimental to a fair assessment of the methods (Wu et al., 2021).

In this article, we aim to present a methodology for a fair and informative comparison between separable ECTS methods in the literature. Specifically, this paper makes the following contributions:

- **A taxonomy** is proposed in Section 2, classifying separable approaches in the literature according to their design choices.

- **Extensive experiments** have been performed, taking into account the obstacles mentioned above. **(1)** The experimental protocol in Section 4.1 explicitly defines the costs used during training and evaluation, and varies the balance between misclassification and delay costs by using a large range of cost values.

**(2)** Experiments are performed repeatedly for several types of cost function, i.e. balanced or unbalanced misclassification costs, and linear or exponential delay costs (see Sections 4.2 and 4.3) and many intermediate results are available in the supplementary materials. **(3)** Ablation and substitution studies are conducted in Section 4.4 with the aim of evaluating the impact of methodological choices, such as the choice of classifier, its calibration, or even z-normalization of training time series, as well as the non-myopia property of some trigger functions. **(4)** The experiments include three baseline approaches, rarely considered in the literature, which often prove to be efficient. **(5)** In addition to the reference data used in the ECTS field, a collection of some thirty non-z-normalized datasets is proposed and provided to the community.

- **An open source library** is made available[1] to enable reproducible experiments, as well as to facilitate the scientific community's development of future approaches. Particular care has been taken to ensure the quality of the code, so that this library may be used to develop real-life applications (see Appendix D).

The scope of this paper is a survey and a benchmark of separable ECTS approaches, in line with our guideline #3. Interested readers can refer to Appendix C for complementary results on end-to-end recent approaches. The rest of this paper is organized as follows: Section 2 proposes and describes an ECTS taxonomy in order to situate separable approaches in relation to each other. It underlines the different choices to be made in a well-founded way when designing an ECTS method. Section 3 presents a review of state-of-the-art methods for ECTS organized along the dimensions of the suggested taxonomy. In Section 4, we present the pipeline developed to conduct extensive experimentation, and we report the main results obtained for different cost settings. Finally, Section 5 concludes this paper.

**Position with respect to other literature surveys** In reaction to the abundant scientific activity around ECTS, Gupta et al. (2020) wrote a survey, however with the following characteristics. First, their perspective on ECTS is confidence-based as they declare "A primary task of an early classification approach is to classify an incomplete time series as soon as possible *with some desired level of accuracy*" (italics are ours) and it is not centered on optimizing the tradeoff between predictive accuracy and earliness. Second, they organize the covered approaches by application domains and methods for representing time series, viz. prefix-based, shapelet-based, model-based, and miscellaneous. A perspective that, in a way, is tangential to the problem of selecting the right decision time.

Another paper (Akasiadis et al., 2024) intends to propose a framework to evaluate ECTS systems. Noticeably, it also adopts a confidence-based approach: "The objective is to find the earliest time-point of a time-series at which a reliable prediction can be made". Alongside the survey, a benchmark is proposed whose characteristics are detailed in Table 3.

The earliest survey published in 2016 by Santos & Kern (2016) deals mostly with classification methods for complete time series. The final section describes four ECTS algorithms with no experiments.

## 2 ECTS: concepts and taxonomy

The aim of this section is (*i*) to formally describe the ECTS problem, and (*ii*) to outline in a principled way the various choices that need to be made when designing one ECTS method.

**Problem statement**

In the ECTS problem, an input time series of size $T$ is progressively observed over time. At time $t \leq T$, the incomplete time series $\mathbf{x}_t = \langle x_1, \dots, x_t \rangle$ is available where $x_{i\,(1 \leq i \leq t)}$ denotes the time-indexed measurements. These measurements can be single or multi-valued. The input time series belongs to an unknown class $y \in \mathcal{Y}$. The task is to make a prediction $\hat{y} \in \mathcal{Y}$ about the class of the incoming time series, at a time $\hat{t} \in [1, T]$ before the deadline $T$.

An ECTS approach aims at optimizing a trade-off between accuracy and earliness of the prediction, and thus must be evaluated on this ground. The correctness of the prediction is measured by the misclassification cost $C_m(\hat{y}|y)$ where $\hat{y}$ is the prediction and $y$ is the true class. The time pressure is sanctioned by a delay cost $C_d(t)$ that is assumed to be positive and, in most applications, an increasing function of time. We thus consider:

---

[1](see https://github.com/ML-EDM/ml_edm)

- $C_m(\hat{y}|y) : \mathcal{Y} \times \mathcal{Y} \to \mathbb{R}$, that corresponds to the cost of predicting $\hat{y}$ when the true class is $y$.

- $C_d(t) : \mathbb{R}^+ \to \mathbb{R}$, the delay cost that, usually, is a non-decreasing function over time.

An ECTS function involves a predictor $\hat{y}(\mathbf{x}_t)$, which predicts the class of an input time series $\mathbf{x}_t$ for any $t \in [1, T]$. The cost incurred when a prediction has been triggered at time $t$ is given by a loss function $\mathcal{L}(\hat{y}(\mathbf{x}_t), y, t) = C_m(\hat{y}(\mathbf{x}_t)|y) + C_d(t)$. The best decision time $t^*$ is given by:

$$t^\star = \underset{t \in [1,T]}{\arg\min} \, \mathcal{L}(\hat{y}(\mathbf{x}_t), y, t). \tag{1}$$

Let $s^\star \in \mathcal{S}$ an optimal ECTS function belonging to a class of functions $\mathcal{S}$, whose output at time $t$ when receiving $\mathbf{x}_t$ is:

$$s^\star(\mathbf{x}_t) = \begin{cases} \emptyset & \text{if extra measures are queried;} \\ y^\star = \hat{y}(x_{t^\star}) & \text{when prediction is triggered at } t = t^\star; \end{cases} \tag{2}$$

ECTS is however an *online* optimization problem, where at each time step $t$ a function $s(\mathbf{x}_t)$ must decide whether to make a prediction or not. Equation 1 is thus no longer operational since it requires complete knowledge of the time series. In practice, the function $s(\mathbf{x}_t)$ triggers a decision at $\hat{t}$, based on a partial description $\mathbf{x}_{\hat{t}}$ of the incoming time series $\mathbf{x}_T$ (with $\hat{t} \leq T$). The goal of an ECTS system is to choose a triggering time $\hat{t}$ as close as possible to the optimal one $t^*$, at least in terms of cost, minimizing $\mathcal{L}(\hat{y}(\mathbf{x}_{\hat{t}}), y, \hat{t}) - \mathcal{L}(\hat{y}(\mathbf{x}_{t^\star}), y, t^\star)$ as much as possible.

From a machine learning point of view, the goal is to find a function $s \in \mathcal{S}$ that best optimizes the loss function $\mathcal{L}$, minimizing the true risk over all time series distributed according to the distribution[2] $\mathbb{P}_{(\mathcal{X} \times \mathcal{Y})}$ that governs the time series in the application:

$$\underset{s \in \mathcal{S}}{\arg\min} \, \mathbb{E}_{(\mathbf{x},y) \sim \mathbb{P}_{(\mathcal{X} \times \mathcal{Y})}} \left[ \mathcal{L}(\hat{y}(\mathbf{x}_{\hat{t}}), y, \hat{t}) \right] \tag{3}$$

In order to solve the ECTS problem, a training set composed of $M$ labeled time series, denoted by $(\mathbf{x}_T^i, y^i)_{i \in [0,M]} \in (\mathcal{X} \times \mathcal{Y})$, where each series $\mathbf{x}_T = \langle x_1, \ldots, x_T \rangle$ is *complete* and of the *same size* $T$, and associated with its label $y \in \mathcal{Y}$, is used to learn how to predict the class of an incoming time series $\mathbf{x}_t = \langle x_1, \ldots, x_t \rangle$, and to learn what can be expected in the future given $\mathbf{x}_t$.

Consequently, model training and deployment are of different natures. The training stage is carried out as a supervised *batch process*, with access to the full labeled time series. When it comes to testing, on the other hand, decision-triggering is an *online process* which stops at time $\hat{t}$, and at the latest, when the deadline $T$ is reached.

## 2.1 Separable vs. end-to-end approaches

An ECTS function must solve both the question of (*i*) *when to stop* receiving new measurements and decide to make a prediction and (*ii*) *how to make the prediction* about the class of the incoming time series $\mathbf{x}_t$.

In the *separable approach*, these questions are solved using two separate components. The *classification* one deals with making a prediction: $\mathbf{x}_t \mapsto \hat{y}$, while the *trigger* function decides when to predict. Within this perspective, the classification component is learned independently of the trigger one, while the latter uses the results of the classification component in order to trigger a decision. We formalize separable approaches by:

$$s(\mathbf{x}_t) = (g \circ h)(\mathbf{x}_t) \tag{4}$$

where $g$ is the decision or trigger function, and $h$ is the prediction function.

In the *end-to-end* approaches, a single component decides when to make a prediction and what that prediction is. Thus, the function $s$, defined in Equation 2, is responsible both for choosing the time $\hat{t}$ for making the predictions, and for the prediction itself $\hat{y}$.

---

[2]Notice that the notation $\mathcal{X}$ is an abuse that we use use to simplify our purpose. In all mathematical rigor, the measurements observed successively constitute a family of time-indexed random variables $\mathbf{x} = (\mathbf{x}_t)_{t \in [1,T]}$. This stochastic process $\mathbf{x}$ is not generated as commonly by a distribution, but by a filtration $\mathbb{F} = (\mathcal{F}_t)_{t \in [1,T]}$ which is defined as a collection of nested $\sigma$-algebras (Klenke, 2013) allowing to consider time dependencies. Therefore, the distribution $\mathcal{X}$ should also be re-written as a filtration.

The question that naturally arises is which type of architecture (i.e. end-to-end or separable) performs best. On the one hand, in separable approaches, the classification component is trained independently of the triggering one, which can be detrimental, for example, by propagating errors from one module to another. On the other hand, separating the ECTS problem into two inherently simpler sub-problems could be an advantage, for example, in terms of convergence during training. Additionally, the separable framework allows one to directly leverage the latest advances in the Time Series Classification (TSC) literature (Bagnall et al., 2017). In this paper, we do not delve any further into the question of which type of architecture is best, which we leave for future work.

The rest of this section is specific to *separable* ECTS approaches, which represent a large part of the literature. In the following, we first examine the decision component (i.e. the trigger function) and then the prediction one (i.e. the classifier). Figure 1 gives a general view of the proposed taxonomy.

## 2.2 Trigger function's properties

The trigger function is responsible for finding when to make a prediction. It triggers the prediction at a time $\hat{t}$ such that $\mathcal{L}(\hat{y}(\mathbf{x}_{\hat{t}}), y, \hat{t})$ is as close as possible to the optimal cost $\mathcal{L}(y^\star(\mathbf{x}_{t^\star}), y, t^\star)$ for the optimal decision time $t^\star$.

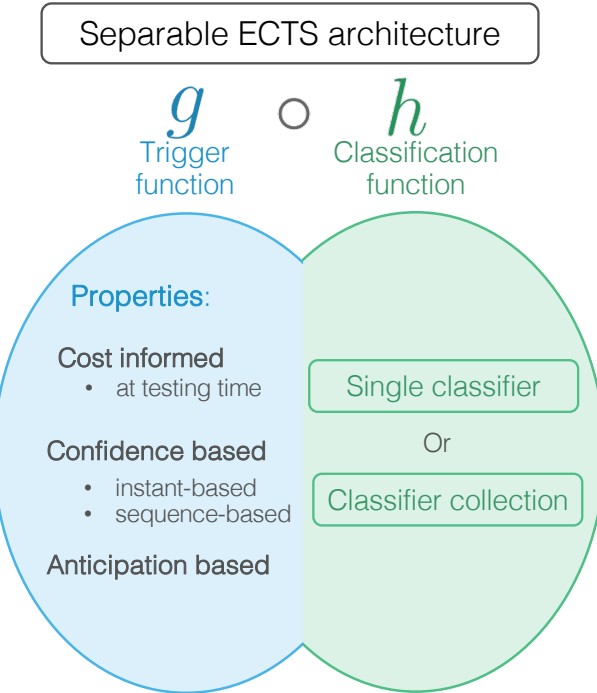

Figure 1: Proposed taxonomy for separable ECTS approaches, which can be viewed as a function composition of trigger and classification functions. The taxonomy does not follow a tree structure as the trigger function may have several properties, that are not mutually exclusive. For instance, a trigger function can be both confidence-based and anticipation based.

### 2.2.1 Cost-informed or cost-uniformed

The first property is whether or not the trigger function actually takes as input any cost functions (or eventually some proxy of them) during training. We call these approaches *cost-informed*. On the contrary, *cost-uninformed* approaches are usually based on some hard pre-defined rules to trigger and are not always easily expandable to a generic cost-setting framework. Additionally, when being *cost-informed*, trigger functions can leverage cost information even more, by being adaptable *at testing time*, i.e. during inference, trigger decisions directly depend on cost functions.

### 2.2.2 Confidence-based approaches

The simplest trigger model of this kind consists in monitoring a quantity related to the confidence of the prediction over time and triggering class prediction as soon as a threshold value is exceeded. The confidence metric monitored can take different forms. For example, a baseline approach, referred to as *Proba Threshold*[3] in the remainder of this paper, involves monitoring $\max_{y \in \mathcal{Y}} p(y|\mathbf{x}_t)$ the highest conditional probability estimated by the classifier. This baseline example is qualified as *instant-based* method, since it takes as input only the last confidence score available at time $t$. Another type of approaches, qualified as *sequence-based*, monitors the entire sequence of past confidence scores, and a triggering prediction is made conditionally on a particular property of this sequence. Accordingly, trigger functions can either take as input a scalar value, e.g. $g(\max_{y \in \mathcal{Y}} p(y|\mathbf{x}_t))$, in the case of *instant-based* approaches, or a sequence of scalar values, e.g. $g(\{\max_{y \in \mathcal{Y}} p(y|\mathbf{x}_\tau)\}_{1 \leq \tau \leq t})$, in the case of *sequence-based* approaches (see Section 3.1).

Those kind of examples can be described as *myopic* since they only look at the current time step $t$ (or past ones as well in the sequence-based case), without trying to anticipate the likely future, i.e. not directly having a decision made based on a forecast of some form. The *anticipation-based* approaches do this.

### 2.2.3 Anticipation-based decisions

As was first noted by Achenchabe et al. (2021), the ECTS problem can be cast as a LUPI (Learning Using Privileged Information) problem (Vapnik & Vashist, 2009). In this scenario, the learner can benefit at the training time from privileged information that will not be available at test time. Formally, the training set can be expressed as $\mathcal{T} = \{(\mathbf{x}_i, \mathbf{x}_i^\star, y_i)\}$, where $\mathbf{x}_i$ is what is observable and $\mathbf{x}_i^\star$ is some additional information not available when the prediction must be made. This is exactly what happens in the ECTS problem. Whereas at test time, only $\mathbf{x}_t$ is available, during training the complete time series is known. This brings the possibility to learn what are the likely futures of an incoming time series $\mathbf{x}_t$ provided it comes from the same distribution. Hence, it becomes also possible to guess the cost to be optimized for all future time steps, and therefore to wait until the moment seems the best. This type of approach can be said *anticipation-based* (also called *non-myopic* in the literature). They can come with many flavors, as long as their decisions are based on some kind of anticipation (see Section 3.2). Because more information from the training set is exploited, it can be expected that these methods outperform *myopic* ones.

Is this confirmed by experience? Are there situations where the advantage is significant? Our experiments in Section 4 provide answers to these questions.

## 2.3 Choice of the classification component

The role of the classification component is to return the prediction $\hat{y}$ of the class of the incoming time series $\mathbf{x}_{\hat{t}}$ at the time decided by the trigger function: $\hat{y} = h_{\hat{t}}(\mathbf{x}_{\hat{t}})$.

One source of difficulty when devising an ECTS method in the separable setting is that, at testing time, inputs differ from one time step to another. When an incoming time series is progressively observed, the number of measurements, and hence the input dimension, varies. Two approaches have been used to deal with the problem.

1. A *set of classifiers* $\{h_t\}_{t \in [1,T]}$ is learned, each dedicated to a given time step $t$, and thus a given input dimension. In practice, authors often choose a limited subset of timestamps, usually a set of twenty (one measurement every 5% of the length of the time series), to restrict the number of classifiers to learn and therefore the associated computational cost.

2. A *single classifier* $h$ is used for all possible incoming time series $\mathbf{x}_t$. One way of doing this is to "project" an input $\mathbf{x}_t$ of dimension $t \times d$, if $d$ is the dimension of an observation at time $t$ (i.e. multi-valued time series), into a fixed dimensional vector whatever $t$ and $d$. This may simply be the mean value and standard deviation of the available measurements (multiplied by the dimension $d$) or the result of a more sophisticated feature engineering as tested by Skakun et al. (2017). Deep learning architectures can also be used to learn an encoding of the time series in an intermediate layer (Wang et al., 2016; Sawada et al., 2022).

---

[3]Baseline implemented in the aeon (Middlehurst et al., 2024a) library : `https://urlz.fr/qmWl`

Table 1: Table of published separable methods for the ECTS problem with their properties along dimensions underlined in the taxonomy. Note that void values indicate that the corresponding property is not present in the referred system.

| References | Classifier(s) (collection ✔) | Confidence | Anticipation | Cost informed |
|---|---|---|---|---|
| Reject (Hatami & Chira, 2013) | SVM | instant | | |
| iHMM (Antonucci et al., 2015) | HMM | instant | | |
| ECDIRE (Mori et al., 2017b) | Gaussian Process (✔) | instant | | |
| Stopping Rule (Mori et al., 2017a) | Gaussian Process (✔) | instant | | ✔ |
| ECEC (Lv et al., 2019) | WEASEL (✔) | sequence | | ✔ |
| TEASER (Schäfer & Leser, 2020) | WEASEL (✔) | sequence | | ✔ |
| SOCN (Lv et al., 2023) | FCN | sequence | | ✔ |
| ECTS (Xing et al., 2012) | 1NN | instant | ✔ | |
| RelClass (Parrish et al., 2013) | QDA, Linear SVM | instant | ✔ | |
| 2step/NoCluster (Tavenard & Malinowski, 2016) | Linear SVM (✔) | | ✔ | ✔(test) |
| ECONOMY-$\gamma$-max (Zafar et al., 2021) | XGBoost + tsfel (✔) | | ✔ | ✔(test) |
| CALIMERA (Bilski & Jastrzebska, 2023) | MiniROCKET (✔) | | ✔ | ✔ |
| FIRMBOUND (Ebihara et al., 2025) | LSTM | | ✔ | ✔ |

Both approaches have their own limitations. On the one hand, using a set of classifiers, each independently dedicated to a time step, does not exploit information sharing. On the other hand, using a single classifier seems to be a more difficult task, as the representation of $\mathbf{x}_t$ can be different at times $t$ and $t+1$ and all further time steps which can lead to additional difficulty for the classifier while moreover requiring a more demanding feature engineering step. Therefore, here also, it is interesting to measure experimentally whether one dominates the other. This will be the subject of future work.

## 3 An organized state-of-the-art of separable methods

In this section, approaches from the literature are considered and organized around two key notions from the introduced taxonomy: *confidence-based* and *anticipation-based*.

Subsection 3.1 presents methods whose decisions are triggered in a myopic way, based on some confidence measure. Subsection 3.2 describes approaches using a non-myopic decision criterion, which attempts to anticipate likely continuations. Table 1 shows how various approaches described in the literature can be organized using the characteristics underlined in the proposed taxonomy.

### 3.1 Confidence-based approaches

There exist two families of confidence-based approaches. In the *first* one, only the last time step is considered, a score based on confidence estimations is monitored at each time step and a class prediction is triggered as soon as a threshold on this score is exceeded. By contrast, in the *second*, a sequence of estimated scores is monitored, and the condition to trigger a decision depends upon some property of this sequence.

#### 3.1.1 Instant-based decision criterion

• One basic method is to monitor $\max_{y \in \mathcal{Y}} p(y|\mathbf{x}_t)$, the highest conditional probability estimated by the classifier, which is a simple measure of classifier confidence over time. As soon as it exceeds a value, which is a hyperparameter of the method, a prediction is made. We call this method PROBA THRESHOLD and use it as a baseline for comparison later in our experiments.

• The REJECT method (Hatami & Chira, 2013) uses ensemble consensus as a confidence measure. For each time step, first (*i*), a pool of classifiers is trained by varying their hyperparameters (i.e. SVMs); then (*ii*), the most accurate of these are selected; and (*iii*) the pair of classifiers minimizing their agreement in predictions is chosen to form the ensemble. Finally, the prediction is triggered as soon as both classifiers in the ensemble predict the same class value.

• Hidden Markov Models (HMMs) are naturally suited to the classification of online sequences. An HMM is learned for each class, and at each time step $t$, the class to be preferred is the one with the highest a posteriori probability given $\mathbf{x}_t$. However, the decision to make a prediction now or to postpone it must then involve a threshold so that the prediction is only made if the a posteriori probability of the best HMM is sufficiently high or is greater than that of the second-best. In reaction to this, Antonucci et al. (2015) propose to replace the standard HMM with *imprecise HMMs* based on the concept of credal classification. This eliminates the need to choose a threshold, since a decision is made when one classification "dominates" (according to a criterion based on probability intervals) all the others.

• Rather than considering only the largest value predicted by the classifier, it is appealing to consider also the difference with the second largest value, since a large difference points to the fact that there is no tie between predictions to expect.

This is one dimension used in the Stopping Rule (SR) approach (Mori et al., 2017a). Specifically, the output of the system is defined as:

$$g(h(\mathbf{x}_t)) = \begin{cases} \emptyset & \text{if extra measures are queried;} \\ \hat{y} = \arg\max_{y \in \mathcal{Y}} p(y|\mathbf{x}_t) & \text{when } \gamma_1\, p_1 + \gamma_2\, p_2 + \gamma_3\, \frac{t}{T} > 0 \end{cases} \tag{5}$$

where $p_1$ is the largest posterior probability $p(y|\mathbf{x}_t)$ estimated by the classifier $h$, $p_2$ is the difference between the two largest posterior probabilities, and $\frac{t}{T}$ represents the proportion of the incoming time series at time $t$. The parameters $\gamma_1$, $\gamma_2$, and $\gamma_3$ are learned from the training set, using genetic algorithm.

• Using the same notations as SR, the Early Classification framework based on class DIscriminativeness and RE-Liability (ECDIRE) (Mori et al., 2017b) finds the earliest timestamp for which a threshold applied on $p_1$ is reached (defined as in Equation 5). Then, the quantity $p_2$ is monitored, and a second threshold is applied to trigger the prediction.

• Ringel et al. (2024) use the *Learning Then Test* (LTT) (Angelopoulos et al., 2021) calibration framework to address ECTS. In practice, the proposed approach greedily computes thresholds at each time step, in order to monitor some conditional control risk measure, given a pre-defined error rate. This paper investigates text applications.

• Historically, a related scenario predates the ECTS problem but is different. In the *sequential decision making* and *optimal statistical decisions* frameworks (DeGroot, 2005; Berger, 1985), the successive measurements are supposed to be independently and identically distributed (i.i.d.) according to a distribution of unknown "parameter" $\theta$. The problem is to determine as soon as possible whether the measurements have been generated by a distribution of parameter $\theta_0$ (hypothesis $H_0$) or of parameter $\theta_1$ (hypothesis $H_1$) with $\theta_0 \neq \theta_1$. In the Wald's Sequential Probability Ratio Test (Wald & Wolfowitz, 1948; Ghosh & Sen, 1991), the log-likelihood ratio $R_t = \log \frac{P(\langle x_1^i, \ldots, x_t^i \rangle \mid y=-1)}{P(\langle x_1^i, \ldots, x_t^i \rangle \mid y=+1)}$ is computed and compared with two thresholds that are set according to the required error of the first kind $\alpha$ (*false positive error*) and error of the second kind $\beta$ (*false negative error*). This beautiful setting allows one to get optimal decision times at the cost of being able to compute the log-likelihood. However, it differs from the ECTS problem, where successive observations are dependent. The i.i.d. assumption being not valid for the ECTS problem, a generalization to the non-i.i.d. case was proposed by Tartakovsky et al. (2014), providing guarantees for the asymptotic case (with $T \to \infty$). Despite this latter limitation, Ebihara et al. (2025) has recently applied this type of approach to ECTS with finite time horizons. The authors propose practical ways of both estimating $R_t$ (Ebihara et al., 2023) and triggering times by solving a backward induction problem (Tartakovsky et al., 2014).

### 3.1.2 Sequence-based decision criterion

Other approaches propose *sequence-based* confidence measures specifically designed for the ECTS problem.

• The Effective Confidence-based Early Classification (ECEC) (Lv et al., 2019) proposes a confidence measure based on the sequence of predicted class values, from the first one observed to the current timestamp. At each time step,

this approach exploits the *precision* of the classifier to estimate the probability for each possible class value $y \in \mathcal{Y}$ of being correct if predicted. Then, assuming that successive class predictions are independent, the proposed confidence measure represents the probability that the last class prediction is correct given the sequence of predicted class values. The proposed confidence measure is monitored over time, and prediction is triggered if this measure exceeds a certain threshold $\gamma$ tuned as the single hyperparameter.

• The TEASER (Two-tier Early and Accurate Series classifiER) (Schäfer & Leser, 2020) approach considers the problem of whether or not a prediction should be triggered as a classification task, the aim of which is to discriminate between *correct* and *bad* class predictions. As the authors point out, the balance of this classification task varies according to the time step considered $t \in [1, T]$. Indeed, assuming there is an information gain over time, there are fewer and fewer bad decisions as new measurements are received (or even no bad decisions after a while, i.e. $\forall\ t > t'\ (0 < t' \leq T)$ for some datasets). To exploit this idea, a collection of one-class SVMs is used, learning hyper-spheres around the correct predictions for each time step. A prediction is triggered when it falls within these hyper-spheres for $\nu$ consecutive time steps.

• The Second-Order Confidence Network approach (SOCN) (Lv et al., 2023) considers, as does TEASER, the same classification task aiming to discriminate between correct and bad predictions. To learn this task, a transformer (Vaswani et al., 2017) is used, taking as input the complete sequence of conditional probabilities estimated by the classifier $h$, from the first time step, up to the current time step. A confidence threshold $\nu$ is learned by minimizing the same cost function as Lv et al. (2019) do, above which the prediction is considered reliable and therefore triggered.

### 3.2 Anticipation-based methods

• One way of designing approaches that anticipate future measurements is to achieve classification of an incomplete time series while guaranteeing a minimum probability threshold, according to which the same decision would be made on the complete series. This is the case of the Reliability Classification (RELCLASS) approach (Parrish et al., 2013). Assuming that the measurements are i.i.d. and generated by a Gaussian process, this approach estimates $p(\mathbf{x}_T | \mathbf{x}_t)$ the conditional probability of the entire time series $\mathbf{x}_T$ given an incomplete realization $\mathbf{x}_t$ and thus derives guarantees of the form:

$$p\big(h_T(\mathbf{x}_T) = y | \mathbf{x}_t\big) = \int_{\mathbf{x}_T\ \text{s.t.}\ h_T(\mathbf{x}_T) = y} p(\mathbf{x}_T | \mathbf{x}_t)\, d\mathbf{x}_T \geq \gamma$$

where $\mathbf{x}_T$ is a random variable associated with the complete time series, $\gamma$ is a confidence threshold, and $h_T$ is the classifier learned over the complete time series. At each time step $t$, $p(h_T(\mathbf{x}_T) = y | \mathbf{x}_t)$ is evaluated and a prediction is triggered if this term becomes greater than the threshold $\gamma$, which is the only hyperparameter to be tuned.

• Another way of implementing anticipation-based approaches is to exploit the continuations of training time series, which are full-length. One of the first methods for ECTS has been derived into such an anticipation-based approach. The first, called Early Classification on Time Series (ECTS) (Xing et al., 2009), exploits the concept of Minimum Prediction Length (MPL), defined as the earliest time step for which the predicted label should not change for the incoming time series $\mathbf{x}_t$ from $t$ to $T$. This is estimated by looking for the 1NN of $\mathbf{x}_t$ in the training set, and checks whether from $t$ onward, its predicted label does not change. To be more robust, the MPL is defined based on clusters computed on full-length training time series to estimate the best decision time. The approach has been extended later on to speed up the learning stage (Xing et al., 2012). This method looks in its own way at the likely future of $\mathbf{x}_t$ - i.e. an incomplete time series belongs to a cluster whose continuations are known - and thus can be considered as an anticipation-based method.

• Dachraoui et al. (2015) present a method that claims explicitly to be "non-myopic" in that a decision is taken at time $t$ only insofar as it seems that no better time for prediction is to be expected in the future. In order to do this, the family of ECONOMY methods estimates the future cost expectation based on the incoming time series $\mathbf{x}_t$. This can be done since the training data consists of full-length time series and therefore a Learning Using Privileged Information (LUPI) (Vapnik & Vashist, 2009) is possible.

More formally, the objective is to trigger a decision when $\mathbb{E}_{y,\hat{y}}[\mathcal{L}(\hat{y}, y, t) | \mathbf{x}_t]$ is minimal, with:

$$\mathbb{E}_{y,\hat{y}}[\mathcal{L}(\hat{y}, y, t) | \mathbf{x}_t] = \sum_{y \in \mathcal{Y}} P(y | \mathbf{x}_t) \sum_{\hat{y} \in \mathcal{Y}} P(\hat{y} | y, \mathbf{x}_t)\, C_m(\hat{y} | y)\ +\ C_d(t) \tag{6}$$

A tractable version of Equation 6 has been proposed by introducing an additional random variable which is the membership of $\mathbf{x}_t$ to the groups of a partition $\mathcal{G}$:

$$\mathbb{E}_{y,\hat{y}}[\mathcal{L}(\hat{y},y,t)|\mathbf{x}_t] = \sum_{g_k \in \mathcal{G}} P(g_k|\mathbf{x}_t) \sum_{y \in \mathcal{Y}} P(y|g_k) \sum_{\hat{y} \in \mathcal{Y}} P(\hat{y}|y,g_k) C_m(\hat{y}|y) + C_d(t) \tag{7}$$

In technical terms, training approaches from the ECONOMY framework involve estimating the three probability terms of Equation 7, for the current time step $t$, as well as for future time steps $t + \tau \in [t+1, T]$, with:

- $P(g_k|\mathbf{x}_t)$ the probability of $\mathbf{x}_t$ belonging to the groups $g_k \in \mathcal{G}$,

- $P(y|g_k)$ the prior probability of classes in each group,

- $P(\hat{y}|y,g_k)$ the probability of predicting $\hat{y}$ when the true class is $y$ within the group $g_k$.

A key challenge in this framework is to design approaches achieving the most *useful partition* for predicting decision costs expectation. In the first article which presents this framework Dachraoui et al. (2015), a method, called ECONOMY-$K$, is designed as follows. (*i*) A partition of training examples is first performed by a K-means algorithm ; (*ii*) then a simple model uses the Euclidean distance as a proxy of the probability that $\mathbf{x}_t$ belongs to each group; (*iii*) the continuation of training time series within each group is exploited to predict the cost expectation for future time steps.

In order to avoid the clustering step with the associated choice of hyperparameters (Tavenard & Malinowski, 2016) presented a variant called NoCLUSTER which uses the 1-nearest neighbor in the training set in order to guess the likely future of $\mathbf{x}_t$.

Then, ECONOMY-$\gamma$ was introduced by Achenchabe et al. (2021) which relies on a supervised method to define a confidence-based partition of training time series. The algorithm, dedicated to binary classification problems, is designed as follows: (*i*) a collection of partitions is constructed by discretizing the output of each classifier $\{h_i\}_{i \in [1,T]}$ into equal-frequency intervals, the groups thus formed correspond to confidence levels for each time step; (*ii*) at the current time $t$, the incoming time series $\mathbf{x}_t$ belongs to only one group, since the output of the classifier $h_t$ falls within a particular confidence level; (*iii*) then, a Markov chain model is trained to estimate the probabilities of the future time step confidence levels. ECONOMY-$\gamma$-MAX (Zafar et al., 2021) generalizes this approach to multi-class problems, aggregating the multiple conditional probabilities in the classifiers' output by using only the most probable class value.

• CALIMERA (Bilski & Jastrzebska, 2023) uses anticipation about the future from another perspective. Instead of trying to guess the likely continuation of $\mathbf{x}_t$ which allows one to compute expected future costs, and therefore to wait until there seems no better time to make a prediction, their method is based on predicting directly the difference in cost between predicting the class now and the best reachable cost from next timestep to last one $T$. If this difference is positive, then it is better to postpone the prediction. They advocate furthermore, that a calibration step should intervene on the collection of classifiers, in order to build meaningful regression target to learn the trigger model.

## 4 Experiments & Results

This section presents the extensive set of experiments carried out in order to provide a consistent and fair evaluation of a wide range of existing literature's methods. We first describe the experimental protocol used. We then turn to the experiments and their results. Figure 2 provides a synthetic view of the organization of these experiments.

- Section 4.1 introduces the experimental protocol as well as the global evaluation methodologies.

- In Section 4.2, eight state-of-the-art methods and the three baseline ones are evaluated using a widely used cost setting, i.e. with a binary balanced misclassification cost and a linear delay cost.

- In Section 4.3, methods are tested in an anomaly detection scenario[4] where the misclassification cost matrix is severely imbalanced, with false negatives being much more costly than false positives, and where the delay cost is no longer linear with time but increases exponentially with time.

- Finally, Section 4.4 briefly describes a set of other experimental setups, derived from either the standard setting or the anomaly detection one, including, for instance, testing the impact of z-normalization. Complementary results can be found in Appendix F.

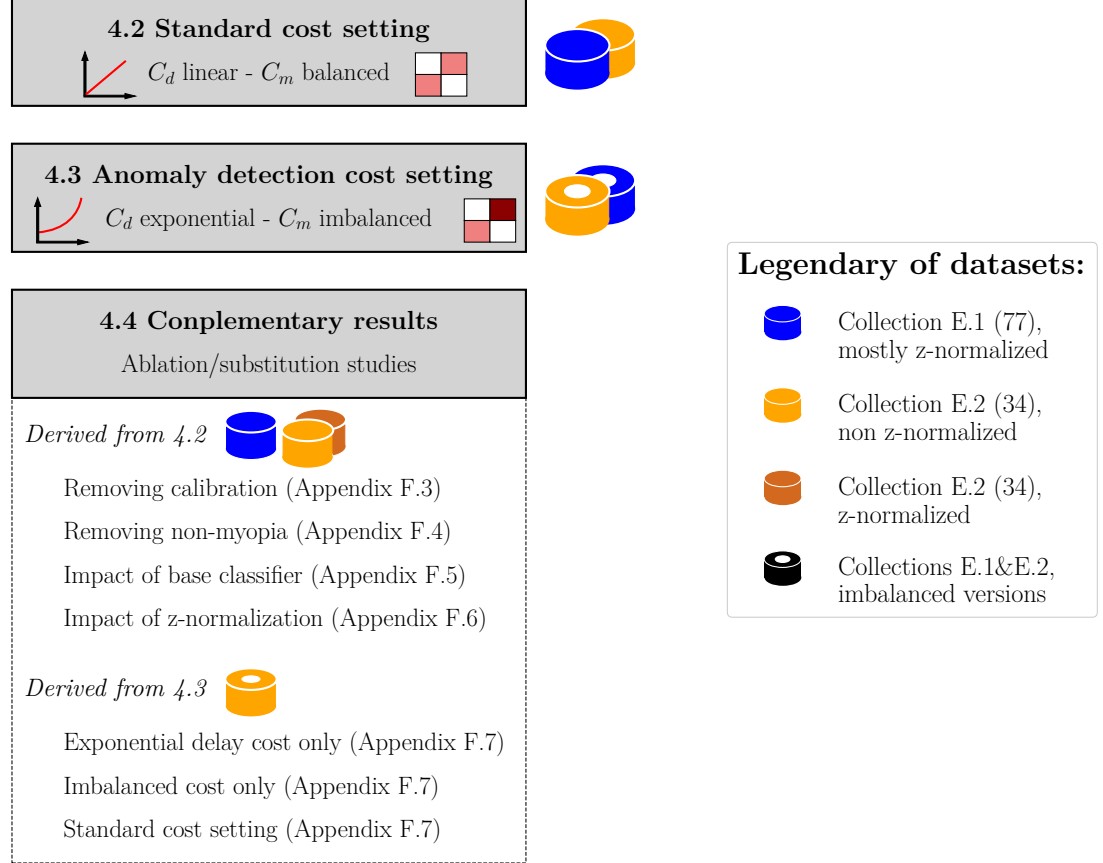

Figure 2: Experiments diagram. While this section mainly discusses results about the cost settings in Subsections 4.2 and 4.3, many other alternative experiments are briefly analyzed in Subsection 4.4 and are more detailed in the Appendix F. Details on the used datasets can be found in Appendix E.

The experiments presented here aim to evaluate the effects of design choices on method performance and thus provide answers to the questions:

- Do *anticipation-based methods* perform better than *myopic* ones?

- Do methods that are *cost-informed* for their decisions (i.e. explicitly estimating costs) perform better than methods that are *cost-uninformed*? (see Section 2.2.1)

- How do the various methods fare when *modifying the form of the delay cost and/or the misclassification cost matrix*?

---

[4]We consider anomalies to be actual phenomena of interest, such as the failure of a machine (Boniol et al., 2024). These anomalies may be accompanied by revealing precursor signals in the time series, which an ECTS system should be able to detect by optimizing both the accuracy of prediction and its earliness.

### 4.1 Experimental Protocol

This section covers the shared part of the experimental protocol for all the experiments, irrespective of the choice of the cost functions (see Sections 4.2 and 4.3 for this). Additional implementation specifications can be found in Appendix B.

#### 4.1.1 Evaluation of the performance

For each test time series $\mathbf{x}^i$, an ECTS method incurs a cost assumed to be of the additive form: $C_m(\hat{y}_i|y_i) + C_d(\hat{t})$, where $\hat{t}$ is the time when the system decided to make a prediction, this prediction being $\hat{y}_i$.

For a test set of $M$ time series, the *average cost* of a method (Equation 8) is used as the criterion with which to evaluate the methods.

$$AvgCost_{\text{test}} \;=\; \frac{1}{M} \sum_{i=1}^{M} C_m(\hat{y}_i|y_i) + C_d(\hat{t}_i) \tag{8}$$

In addition, in order to assess how the methods adapt to various balances between the misclassification and the delay costs, we vary the settings of these costs by weighting them during training and testing. The performance of the methods is therefore evaluated using the weighted average cost, as defined in Equation 9, for different values of the costs balance $\alpha$, ranging from 0 to 1, with a 0.1 step:

$$AvgCost_\alpha = \frac{1}{M} \sum_{i=0}^{M} \alpha \times C_m(\hat{y}_i|y_i) + (1-\alpha) \times C_d(\hat{t}_i) \tag{9}$$

*Small values* of $\alpha$ correspond to a high delay cost and a small misclassification cost ; inversely, *large values* of $\alpha$ give more weight to the misclassification cost with a lower delay cost.

#### 4.1.2 Comparing the trigger methods

Our experiments aim foremost at comparing the trigger methods used that are responsible for deciding when to make a prediction about the class of the incoming time series. To this end, all compared methods in our experiments use the same prediction component. As advocated by Bilski & Jastrzebska (2023), we have chosen the MINIROCKET algorithm (Dempster et al., 2021) to be the base classifier for all methods. It is indeed recognized as among the best performing classifiers in the time series classification literature as well as one of the fastest ones. However, we have carried out additional experiments with two other classifiers (see Section 4.4). In our experiments, we have additionally reported results for *end-to-end* methods in order to situate the performance of these methods in respect to separable ones in Appendix C.2.

**Trigger models**: Eight trigger models were selected from the literature based on their usage and their performances[5]: ECONOMY-$\gamma$-MAX (Achenchabe et al., 2021), CALIMERA (Bilski & Jastrzebska, 2023), STOPPING RULE (Mori et al., 2017a), TEASER$_{HM}$ (Schäfer & Leser, 2020), TEASER$_{Avg}$, ECEC (Lv et al., 2019), ECDIRE (Mori et al., 2017b), ECTS (He et al., 2013).

All these methods have been re-implemented using Python, reproducing results close to the published ones. Except for the code for the ECTS implementation, which has been taken from Kladis et al. (2021). Hyperparameters are the ones chosen in the original published methods. Code to reproduce the experiments is available publicly at (see https://github.com/ML-EDM/ECTS_survey) .

**Baselines**: Furthermore, in order to evaluate the benefits, if any, of the various methods, it is telltale to compare them with simple ones. We chose three such baselines:

- ASAP (As Soon As Possible) always triggers a prediction at the first possible timestep.

---

[5]The *EDSC* algorithm (Xing et al., 2011), even though available in the provided library, is not included in the following experiments, due to high space and time complexity (which hinders fair comparisons.)

- ALAP (As Late As Possible) always waits for the complete series to trigger the prediction.

- PROBA THRESHOLD is a natural, confidence-based, cost-informed, baseline: it triggers a prediction if the estimated probability of the likeliest prediction exceeds some threshold, found by grid search (cf. Section 3.1, Confidence-based).

### 4.1.3 Datasets and training protocol

**Two collections of datasets**[6]**:** In order to be able to directly compare our results to past experiments, we first use the usual TSC datasets from the UCR Archive (Dau et al., 2019) with the default split. In total, we have used 77 datasets from the UCR Archive, i.e. the ones with enough training samples to satisfy our experimental protocol from the start to the end, i.e. with at least one example per class within each of the used disjoint subsets (see *Splitting strategy* below). (blue cylinder in Figure 2). In this way, most of the datasets used by either Mori et al. (2017a) and Lv et al. (2019), or by Achenchabe et al. (2021) are contained in our experiments.

A second collection of non z-normalized datasets is also provided. In this way, the associated potential information leakage is avoided (see Section 1). Any difference in the performance obtained on the z-normalized data sets can thus signal the danger of z-normalization with firm evidence. Considering the limited amount of non z-normalized datasets within the UCR archive (Dau et al., 2019), we have decided to look for complementary new datasets so as to provide another collection of datasets. To this end, the Monash archive for extrinsic regression (Tan et al., 2020), provided 20 new time series datasets, for which we have discretized the numeric target variable into binary classes based on a threshold value. For instance, if this threshold is equal to the median value of the regression target, the resulting classification datasets will be balanced in terms of classes (as in Section 4.2). Note that this threshold can be chosen differently to get imbalanced datasets (as in Section 4.3.2), several thresholds could also be used to increase the number of classes. As a result, we obtain a new set of classification tasks, as has recently been done by Middlehurst et al. (2024b). In the end, 34 datasets have been gathered: 14 from the original archive and 20 from the Monash extrinsic regression archive. (orange cylinder in Figure 2). More details about the selection of datasets can be found in Appendix B.4

**Splitting strategy:** When not using predefined splits, the train sets are split into two distinct sets in a stratified fashion: a first one to train the different classifiers, corresponding to 40% of the training set and another one to train the trigger model, trained over the 60% left. The set used to train the classifiers is itself split into two different sets in order to train calibrators, using 30% of the given data. Because of this procedure, we have been led to exclude some of the datasets, due to their limited training set size.

All the experiments have been performed using a Linux operating system, with an Intel Xeon E5-2650 2.20GHz (24-cores) and 252GB of RAM. Proceeding all datasets (including both blue and orange cylinders) over all competing approaches takes between 9-10 days, using MINIROCKET classifier, which is the most efficient tested.

## 4.2 Experiments with balanced misclassification and linear delay costs

This first setting is the one most widely used in the literature to date.

### 4.2.1 Cost definition

The misclassification cost is symmetrical and balanced; the delay cost is linear. They can be defined as follows:

$$C_m(\hat{y}|y) = \mathbb{1}(\hat{y} \neq y)$$
$$C_d(t) = \frac{t}{T}$$

---

[6]All original datasets of the paper can be downloaded, already prepared and split, from `https://urlz.fr/qRqu`

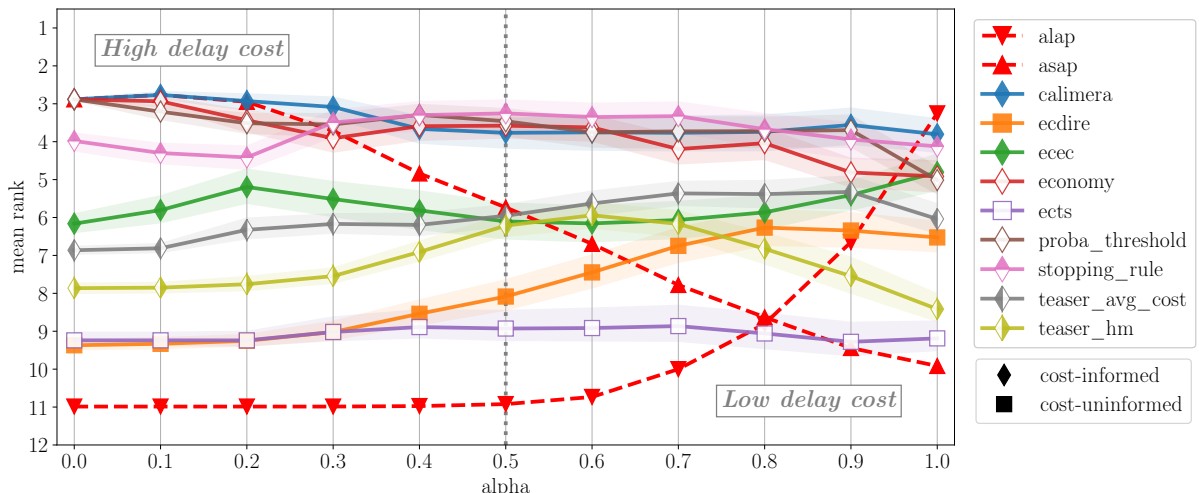

(a) Evolution of the mean ranks, for every $\alpha$, based on the *AvgCost* metric. Shaded areas correspond to 90% confidence intervals.

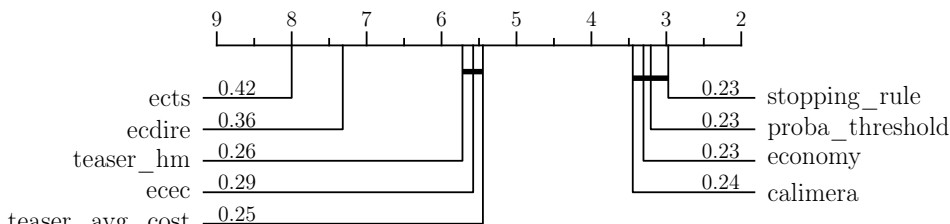

(b) Alpha is now fixed to $\alpha = 0.5$. Wilcoxon signed-rank test labeled with mean *AvgCost*.

Figure 3: The ranking plot (a) shows that, across all values of $\alpha$, a top group of four approaches distinguishes itself. The significance of this result is supported by statistical tests. Specifically, we report this for $\alpha = 0.5$ as shown in (b).

Thus, for each dataset, the $AvgCost_\alpha$ is bounded between 0 and 1, as, within this cost definition, the average misclassification (resp. temporal) cost, across examples is equivalent to the $1 - Accuracy$ (resp. *Earliness*) measure, with $Accuracy = \frac{1}{M} \sum_{i=1}^{M} \mathbb{1}(\hat{y}_i = y_i)$ and $Earliness = \frac{1}{M \times T} \sum_{i=1}^{M} \hat{t}_i$.

### 4.2.2 Results and analysis

For comparability reasons, this first set of experiments is analyzed over the classical ECTS benchmark used in the literature so far (blue cylinder in Figure 2). Results over the new, non z-normalized, datasets can be found in Appendix F.

Figure 3a allows for a broad look, varying the relative costs of misclassification and delaying prediction using Equation 9, where a small value of $\alpha$ means that delay cost is paramount. 90% level confidence intervals have been computed using bootstrap[7]. Again, the same four methods top the others for almost every value of $\alpha$. Not surprisingly, the baseline ASAP (predict as soon as possible) is very good when the delay cost is very high, while ALAP (predict at time $T$) is very good when there is no cost associated with delaying decision.

When evaluated by their *average rank* on all data sets with respect to the average cost (Equation 9), here for $\alpha = 0.5$, four methods significantly outperform the others:

---

[7]Resample with replacement has been done a large number of times ($10.000\times$) and are reported as shaded colors in the figure. The statistic of interest is studied, here the mean, by examining the bootstrap distribution at the desired confidence level.

Table 2: Leading ECTS methods and their properties along dimensions underlined in the taxonomy.

| Methods | Confidence | Anticipation | Cost-informed |
|---|---|---|---|
| STOPPING RULE | ✓ | | ✓ |
| PROBA THRESHOLD | ✓ | | ✓ |
| ECONOMY-$\gamma$-MAX | | ✓ | ✓(test) |
| CALIMERA | | ✓ | ✓ |

Figure 3b provides a view about the relative performances of the tested methods in terms of the average cost induced using the methods for $\alpha = 0.5$. The Wilcoxon-Holm Ranked test provides an overall statistical analysis. It examines the critical difference among all techniques to plot the method's average rank in a horizontal bar. Lower ranks denote better performance, and the methods connected by a horizontal bar are similar in terms of statistical significance.

It is remarkable that, in this cost setting, the simple PROBA THRESHOLD method exhibits a strong performance for almost all values of $\alpha$. It is therefore worth including in the evaluation of new methods. However, while Figures 3a, 3b are useful for general analysis, they do not provide insights about how the Accuracy vs. Earliness trade-off is optimized for each of the competitors. Figure 4 provides some explanation for this.

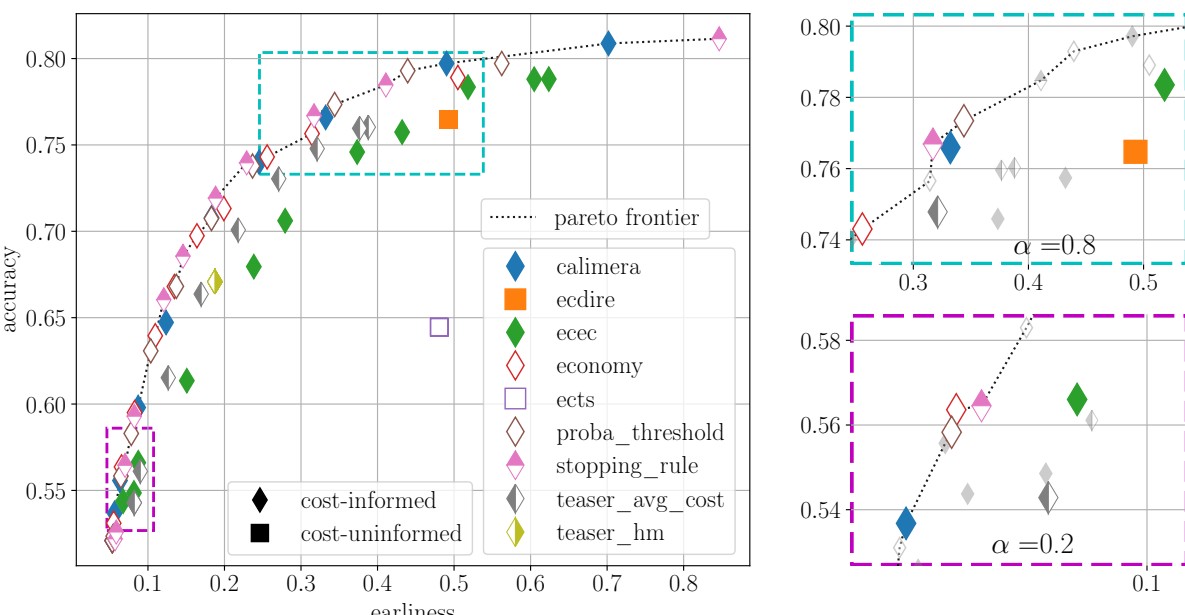

Figure 4: Pareto front, displaying for each $\alpha$ the *Accuracy* on the *y*-axis and *Earliness* on the *x*-axis. Best approaches are located on the top left corner. In zoomed boxes, on the right of the Figure, points corresponding to a single $\alpha$ are highlighted, while other points are smaller and gray. Each of the trigger model is optimizing the trade-off in its own way, resulting in many different approaches having points in the Pareto dominant set.

In this figure, the two evaluation measures: *Accuracy* and *Earliness*, are considered as dimensions in conflict. The *Pareto front* is the set of points for which no other point dominates with respect to both *Accuracy* and *Earliness*. It is drawn here when varying their relative importance using $\alpha$ (in the set $\{0, 0.1, 0.2, \ldots, 1.0\}$).

One must note first that, as ECTS and ECDIRE are cost-uninformed, their performance does not vary with $\alpha$. Whatever the relative weight between accuracy and earliness, they make their prediction approximately after having observed half of the time series and they reach an average accuracy respectively near 0.64 and 0.77. They are clearly dominated by the other methods. This is also the case for TEASER$_{HM}$, which, while being *cost-informed*, only appears once in the figure. Indeed, no weighting mechanism is provided in the original version of the algorithm, where the harmonic mean is used as an optimization criterion (Schäfer & Leser, 2020).

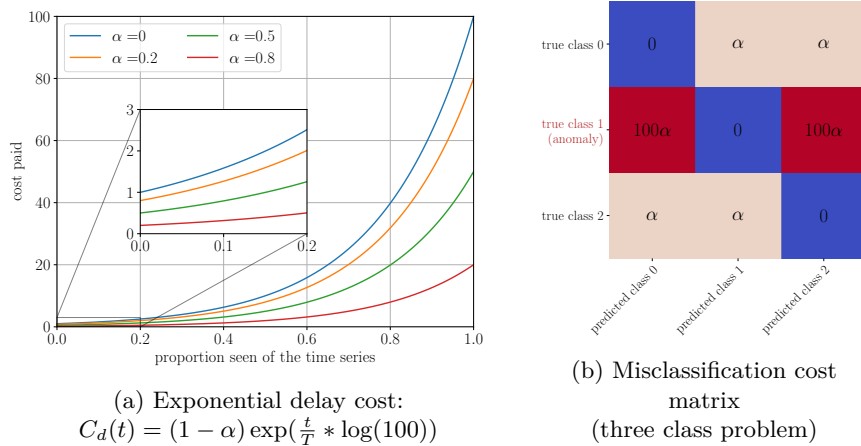

(a) Exponential delay cost:
$$C_d(t) = (1 - \alpha) \exp(\tfrac{t}{T} * \log(100))$$

(b) Misclassification cost
matrix
(three class problem)

Figure 5: Representative delay cost (a) and misclassification ones (b) for an anomaly detection scenario. In our experiments, $\alpha \in [0, 1]$.

Each of the leading methods STOPPING RULE, PROBA THRESHOLD, ECONOMY and CALIMERA have at least one point on the Pareto front and generally exhibit combined performance very close to it. A closer look reveals how each approach optimizes the *Earliness* vs. *Accuracy* trade-off differently for a fixed cost. If we consider $\alpha = 0.8$, for example, it appears that ECONOMY takes its decision earlier than PROBA THRESHOLD, itself being more precocious than ECEC. Because this is also an area of problems where the delay cost is low, by doing so, ECONOMY prevents itself from benefiting from waiting for more measurements and increasing its performance. Hence its slight downward slope on Figure 3a for high values of $\alpha$.

It is worth noting that the two naive baselines ASAP and ALAP perform better than the majority of approaches on seven $\alpha$ values out of ten. This is especially the case when the delay cost is large, i.e. for $\alpha \in [0.1, 0.3]$, for which the ASAP baseline is as competitive as top performers. Globally, the performance of PROBA THRESHOLD is remarkable in this cost setting. Even though it is simply based on a single threshold on the confidence in the current prediction, its performance makes it one of the best methods.

The results computed over the proposed datasets ensemble (i.e. orange cylinder) are displayed in Figure 10 of Appendix F. No significant changes can be observed in the ranking of competing approaches.

## 4.3 Experiments with *unbalanced* misclassification and *non-linear* delay costs

While the previous section has provided a first assessment of how the various methods adapt to different respective weights for the misclassification and the delay costs, it nonetheless assumed that the misclassification costs were balanced (e.g. 0 if correctly classified and 1 otherwise) and that the delay cost was a linear function of time.

There are, however, applications where these assumptions do not hold, for instance, predictive maintenance or hospital emergency services, are characterized by (*i*) *unbalanced* misclassification costs (e.g. it is more costly to have to repair a machine than to carry out a maintenance operation that turns out not being necessary) and by (*ii*) *non-linear* delay costs (e.g. usually, the later the surgical operation is decided, the costlier it is to organize it and the larger the risk for the patient). In the following, we call all applications presenting these characteristics "anomaly detection" applications.

The question arises as to how the various ECTS algorithms behave in this case, depending on their level of cost awareness and whether or not they are anticipation-based. This is what is investigated in the series of experiments reported in this section.

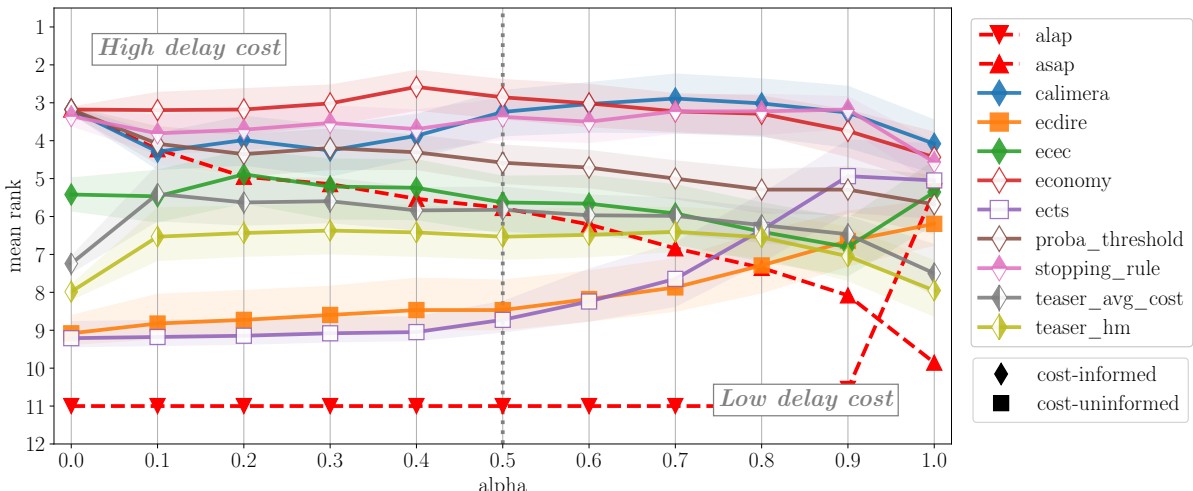

(a) Evolution of the mean ranks, for every $\alpha$, based on the *AvgCost* metric. Shaded areas correspond to 90% confidence intervals.

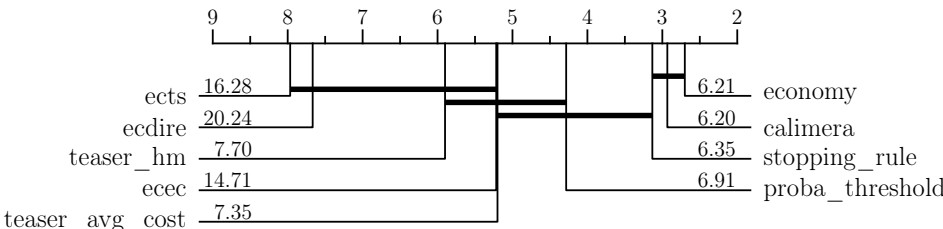

(b) Alpha is now fixed to $\alpha = 0.5$. Wilcoxon signed-rank test labeled with mean *AvgCost*.

Figure 6: The ranking plot (a) shows that, across all $\alpha$, a top group composed by three approaches distinguish. This result is significant as supported by statistical tests. Specifically, for $\alpha = 0.5$ as shown in (b).

### 4.3.1 Cost definition for anomaly detection

In order to study the behavior of the various algorithms on scenarios corresponding to anomaly detection, we set the *unbalanced misclassification cost* matrix such that a false negative (i.e. missing an anomaly) was 100 times costlier than a false positive (i.e. wrongly predicting an anomaly) (see Figure 5b). For this last situation, the delay cost was arbitrarily set to 1. The *delay cost* is defined as an exponential function of time. In order to have a delay cost commensurable with the misclassification one, we decided that waiting for the entire time series to be seen, at $T$, would cost $100 \times \alpha$ (see Figure 5a), starting at $(1 - \alpha)$ for $t = 0$ and reaching $100 \times \alpha$ when $t = T$.

### 4.3.2 Results and analysis

In this part, as a new cost setting is explored, there is no need to produce comparable results from previous works. Thus, we choose to use the new non z-normalized datasets collection (orange cylinder in Figure 2). In order for the imbalanced misclassification cost to make sense, the datasets have been altered so that the minority class represents 20% of all labels. As explained in Section 4.1, some extrinsic regression datasets are turned into classification ones. In these cases, the threshold value has been set to the second decile of the regression target. For the original classification datasets, the minority class has been sub-sampled when necessary.

Results from the Wilcoxon-Holm Ranked test (both regarding the average rank and the value for *AvgCost*) (see Figure 6b) and from the *AvgCost* plot (see Figure 6a) with varying values of $\alpha$ (in Equation 9) show that now, the best method overall is ECONOMY which is cost-informed at testing time, in addition to being anticipation-based. However,

STOPPING-RULE is a very strong contender while being cost-informed but not at testing time and confidence-based. There is a reason for it. When STOPPING RULE equals or overpasses ECONOMY, this applies to high values of $\alpha$ when the delay cost loses its importance, therefore leaving the misclassification cost to reign and confidence-based methods to be effective.

It may come as a surprise that CALIMERA lags behind ECONOMY for $\alpha \in [0, 0.4]$, despite being similarly based on the estimation of future cost expectations. One reason for this is that the cost expectation is achieved by considering only the predicted class. This poor estimate of the cost expectancy becomes critical when the delay cost is important.

Similarly, PROBA THRESHOLD is surprisingly good in this scenario, even if it is no longer in the top tier. Looking solely at prediction confidence, we might expect it to be blind to the rapid increase in delay cost in the anomaly detection scenario. However, it is noticeable that the cost of delay only increases sharply after around 60% of the complete time series has been observed, which is generally sufficient to exceed the confidence threshold. Hence, PROBA THRESHOLD does not suffer from high delay costs that are to come, and exhibits good performance here.

Figure 7 plots the Pareto front, considering two axes based on decision costs. The horizontal axis corresponds to the average delay cost incurred for each example, normalized by the worst delay cost paid at $t = T$. It is better to be on the left of the $x$-axis. The vertical axis corresponds to one minus the misclassification cost incurred for each example, normalized by the worst prediction cost. It is better to be high on the $y$-axis.

We observe that the Pareto front is composed almost exclusively of points corresponding to the ECONOMY and CALIMERA methods. This is consistent with the evaluation based on the *AvgCost* metric. This figure highlights the fact that the design of approaches capable of handling arbitrarily parameterized decision costs requires a cost-informed application framework.

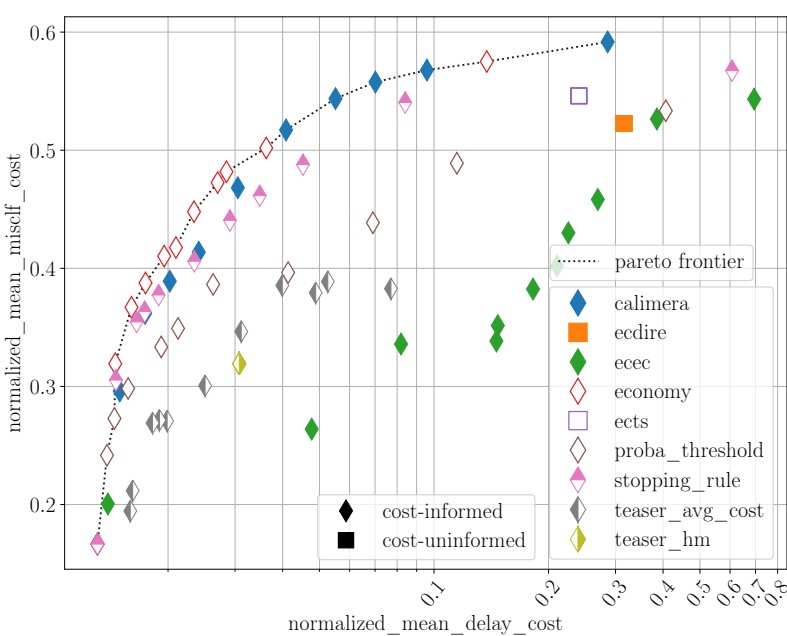

Figure 7: Pareto front, displaying for each $\alpha$, the normalized version of the *AvgCost*, decomposed over delay and misclassification cost on $x$-axis and $y$-axis respectively. Best approaches are located on the top left corner. Due to the exponential shape of the delay cost, the $x$-axis is on log scale.

## 4.4 Other experiments: ablation and substitution studies

In this section, complementary experiments, namely ablation studies as well as sanity checks are briefly discussed. For the sake of brevity, the figures supporting the analysis are reported in Appendix F.

**Impact of removing calibration**

Bilski & Jastrzebska (2023) assert that calibration of the classifiers is paramount for the performance of ECTS algorithms. In order to test this claim, we have repeated the experiment, removing the calibration step. The examples used for calibration have also been removed during training, so that all else remains the same as before.

The results of Figure 13 in Appendix F.3 show that indeed CALIMERA suffers greatly if no calibration is done. Indeed, this approach relies on estimating the expectation of future costs via a regression problem, and miscalibration may have a negative impact on the built targets. For its part, PROBA THRESHOLD suffers somewhat mildly. This is no surprise, as they rely on a single threshold on the confidence of the prediction for all time steps.

**Impact of removing non-myopia**

Experiments have shown that *anticipation-based* approaches tend to outperform the myopic competitors; we further validate this by conducting an ablation study over the *non-myopia* of both ECONOMY and CALIMERA.

Figure 14 in Appendix F.4 clearly shows that, for weak temporal costs, the *anticipation-based* property is critical, whereas *myopic* counterparts significantly fall behind original methods. Intuitively, for higher temporal costs, the *anticipation-based* property is less important as decisions tend to be taken earlier, and it is less necessary for the methods to anticipate the future.

**Impact of the choice of base classifier**

All methods have been compared using the same classifier: MINIROCKET so that only the decision components differ. However, the choice of the base classifier could induce a bias favoring or hampering some methods. In order to clarify this, we have repeated the experiments replacing MINIROCKET with two base classifiers: WEASEL 2.0 (Schäfer & Leser, 2023), and the XGBOOST classifier (Chen et al., 2015) using features produced by TSFRESH (Christ et al., 2018). Both of these classifiers have already been tested within the ECTS literature by Schäfer & Leser (2020); Lv et al. (2019) and Achenchabe et al. (2021) respectively. Figure 15 and 16 in Appendix F.5 report the results respectively with these two classification methods. One can observe that the results are not significantly altered with the same overall ordering of the methods when varying the value of $\alpha$. Furthermore, our results on *AvgCost* show that performance tends to be better for all methods using MINIROCKET (see Table 8 in Appendix F.5). It is thus to be preferred given its simplicity and good performance.

**Impact of z-normalization**

Considering the newly proposed ensemble of datasets, we were not able to identify any problems of information leakage over time. This inconclusive result simply indicates that the variance of the time series measurements is not informative for these datasets, which still could be the case considering past published results. For further details, please refer to Appendix F.6.

## 5 Conclusion

In this paper, we have proposed a **taxonomy** that allows to underline the main families of separable approaches for the ECTS problem, pointing out the essential components and the questions that have to be tackled when designing a new method. We have thus enlightened (*i*) the importance of the two components: *decision* and *prediction*, (*ii*) the distinction between *anticipation-based* and *myopic* methods, and (*iii*) between *cost-informed* and *cost-uninformed* techniques.

We have defined a **methodology for evaluating and comparing ECTS methods**. In addition, and we hope this will prove useful to the community, we have built an **open source library** that includes systematic implementation of the methods tested, the proposed evaluation protocol, as well as a collection of 34 datasets designed to enable informative testing of the methods.

We have also underlined the **importance of considering a variety of cost settings** in the evaluation of ECTS methods so as to reflect what can happen in real-life applications.

The **in-depth experiments** carried out shed light on design choices. Firstly, *are anticipation-based methods better than myopic ones*? We have shown that four of the eleven methods tested perform significantly better. However, these methods do not share the same characteristics with regard to this question. Secondly, to the question: *do cost-informed methods outperform cost-uninformed methods*? The Pareto front indicates a clear superiority of the former. Finally, when we test the *robustness of methods to variations in cost functions*, anticipation-based methods prove superior and, among them, cost-based methods fare even better.

Our experiments have also shown that *calibration* of the classifiers has a large impact on some methods (e.g. CALIMERA) in particular, less so on other methods (e.g. PROBA THRESHOLD and ECDIRE). As regards to the z-normalization of time series which would be unwisely used in evaluation studies, our results (see Appendix F.6) show that the impact on the performance is limited, thus suggesting that the existing trigger functions do not or only slightly benefit from this information leakage.

**Future work** could be carried out to study the literature's approaches applied in as yet *unexplored cost settings*. For example, in many applications, the delay cost depends on the true class and the predicted one, and thus a single cost function integrating misclassification and delay costs should then be used. This general cost form requires the adaptation of some state-of-the-art methods and has not yet been studied.

In addition, in real ECTS applications, it is up to the business expert to define the costs, which is not an easy task in practice. Among the challenges, applications where the costs actually paid are not deterministic are of key interest (e.g. a manufacturing defect on an engine part does not necessarily lead to a failure, but it does increase the probability of paying a higher cost). Thus, future work could study the impact of *stochastic cost functions*. Another interesting case is applications where the costs paid are slightly different, or changed, from those defined by the business experts for the training phase (e.g. a change in the price of raw materials). Those kinds of *cost drift* between training and testing stages could also be further studied.

Finally, in the case of existing separable approaches, the misclassification cost is not exploited for training the classification function. Future work could investigate the interest of using cost-sensitive classifiers in the case of ECTS.

Beyond separable approaches, a more extensive review of end-to-end ECTS methods could be interesting to complete the one presented in this paper. In particular, an experimental protocol should be designed to place end-to-end and separable methods on similar ground and to design new, efficient approaches.

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

# A  Benchmarking: related studies

Table 3: Table of published methods for the ECTS problem with their benchmarking protocol.

| References | No. of datasets | No. of methods | Stats test | Eval. metric(s) | Cost balance | Code |
|---|---|---|---|---|---|---|
| Reject (Hatami & Chira, 2013) | 1 | 1 | | Acc./Earl. | fixed | |
| iHMM (Antonucci et al., 2015) | 1 | 2 | | $\ell_{0/1}$ | fixed | |
| ECDIRE (Mori et al., 2017b) | 45 | 5 | ✓ | Acc./Earl. | fixed | ✓ |
| Stopping Rule (Mori et al., 2017a) | 45 | 5 | ✓ | Acc./Earl. | vary | ✓ |
| ECEC (Lv et al., 2019) | 45+8 | 6 | ✓ | Acc./Earl. | vary | ✓ |
| TEASER (Schäfer & Leser, 2020) | 45 | 5 | ✓ | Acc./Earl./HM | vary | ✓ |
| SOCN (Lv et al., 2023) | 45 | 10 | ✓ | Acc./Earl./HM | fixed | ✓ |
| ECTS (Xing et al., 2012) | 23 | 3 | | Acc./Earl. | fixed | |
| RelClass (Parrish et al., 2013) | 15 | 2 | | Acc./Earl. | vary | ✓ |
| 2step/NoCluster (Tavenard & Malinowski, 2016) | 76 | 3 | ✓ | $AvgCost$/Acc. | vary | ✓ |
| ECONOMY-$\gamma$-max (Zafar et al., 2021) | 45/34 | 2 | ✓ | $AvgCost$ | vary | ✓ |
| CALIMERA (Bilski & Jastrzebska, 2023) | 45/34 | 8 | ✓ | $AvgCost$/Acc./Earl./HM | fixed | ✓ |
| FIRMBOUND (Ebihara et al., 2025) | 7 | 5 | | AAPR/SAT | vary | ✓ |
| EDSC (Xing et al., 2011) | 7 | 4 | | Acc./Earl. | fixed | ✓ |
| EARLIEST (Hartvigsen et al., 2019) | 4 | 3 | | Acc./Earl. | vary | ✓ |
| DDQN (Martinez et al., 2020) | 1 | 2 | | Acc./Earl. | vary | |
| DETSCNet (Chen et al., 2022) | 12 | 5 | | HM | fixed | |
| Benefitter (Shekhar et al., 2023) | 14 | 7 | | Acc./Earl. | vary | ✓ |
| CIS (Cao et al., 2023) | 3 | 3 | | Acc./Earl. | fixed | ✓ |
| ELECTS (Rußwurm et al., 2023) | 4 | 2 | | Acc./Earl. | vary | ✓ |
| EarlyStop-RL (Wang et al., 2024) | 1 | 4 | | FPR/FNR/Earl./F1 | fixed | ✓ |
| Empirical Survey (Akasiadis et al., 2024) | 3 | 5 | | Acc./Earl./HM/F1 | fixed | ✓ |

Columns are defined as :

- *No. of datasets*: the number of datasets used in the paper.

- *No. of methods*: the number of methods tested in the paper (including the one proposed, if any).

- *Stats test*: whether evaluation has been performed using statistical tests (Rainio et al., 2024).

- *Eval. metric(s)*: evaluation metrics used.

- *Cost balance*: whether experiments have been conducted for different cost balances between $C_m$ and $C_d$.

- *Code*: is the code of the method available.

Experimental methodology is heterogeneous in the literature so far, making comparisons between methods more difficult. When considering datasets used or evaluation metrics, a variety of different approaches can be found, as shown in Table 3.

# B  Experimental protocol

## B.1  Optimization of the parameters of the methods

Eight methods from the literature have been tested, respecting as far as possible the choices made in the original papers. Two groups of hyperparameters need to be set: ($i$) some of them are meta parameters independent of the

dataset and have been fixed according to the original papers, (*ii*) others have to be optimized using a grid search based on the *AvgCost* criterion. The optimization of the second group of hyperparameters has been carried out using the value bounds mentioned in the originally published papers. When possible, the granularity of the grid has been adapted to keep similar computation times between competitors. These two groups of hyperparameters are described for all methods in the Appendix B.2. As a remark, because the original version of TEASER uses the harmonic mean (Schäfer & Leser, 2020), we have kept this setting (the resulting method being TEASER$_{HM}$), and we have added a variant called TEASER$_{Avg}$ optimized using *AvgCost*.

## B.2 Hyperparameters

Table 4: Hyperparameters' value. $\Delta$ defines grid's size when performing grid-search over continuous valued intervals.

| Method | Hyperparameters | |
| --- | --- | --- |
| | Fixed | Optimized |
| CALIMERA | *kernel*: "rbf" | |
| ECDIRE | *perc_acc* $= 100\%$ | |
| ECTS | *support* $= 0$ | |
| ECONOMY | | $k \in [\![1 .. 20]\!]$ |
| PROBA THRESHOLD | | $\Delta = 40$ |
| STOPPING RULE | | $\gamma_1, \gamma_2, \gamma_3 \in [-1, 1]$ $\Delta = 10^3$ |
| TEASER_$*$ | | $\nu \in [\![1 .. 5]\!]$ |

## B.3 Calibration of the classifications

Like Bilski & Jastrzebska (2023), we add a calibration step when learning the classifiers, i.e. Platt's scaling (Platt et al., 1999). Indeed, as we are dealing with collections of independently trained classifiers, the prediction scores may not remain consistent with one another over the time dimension. However, the trigger methods usually have their parameters set with the same values for all time steps. This is the case, for example with the PROBA THRESHOLD approach. In addition, some approaches such as CALIMERA and ECONOMY-$\gamma$-MAX exploit the estimated posterior probabilities $\{p(y|\mathbf{x}_t)\}_{y \in \mathcal{Y}}$ to estimate the future cost expectation. It is therefore highly desirable for all classifiers, at all times, to have their output calibrated and is necessary for a fair comparison.

## B.4 Datasets selection

When a time series does not bring an increasing level of information over time about its class, classifiers are likely to obtain the same confusion matrix for all time steps, and the trigger function should then choose to make a prediction at the first time step to avoid the delay cost even if the prediction is very uncertain. Although this case may occur, it cannot form the basis for comparing the trigger functions. This is why we have been careful about the selection of data sets.

We have chosen data sets where the classification performance tends to increase over time, as measured with the same basis classifier for all methods (see Section 4.1.2). We check that the train AUC increases averaged over a time window of size 5, when using either the beginning of the series, or almost complete ones. Specifically, the beginning of the series is taken using $5\%, \ldots, 25\%$, and the almost complete series are of the following lengths $75\%, \ldots, 100\%$. When the difference of averaged AUC is strictly positive, the dataset is selected. (see Table 7). In addition, all datasets exhibiting z-normalization have been removed. Thirty-four datasets passed these demanding criteria (see Appendix E).

For instance, Figure 8 (left) displays the time series of a synthetic data set where each of the three classes is characterized by a specific pattern over time. Class 0 can thus be recognized early on, while class 1 and class 2 can only be discriminated after approximately one third of the length of the time series. This is confirmed by the accuracy of the recognition of each class (right).

Selecting data sets appropriate for the comparison of ECTS methods is a demanding process that must check for all the desirable properties described above. We thus hope that the corpus thus built and made available on the open source library will be useful to the scientific community for future performance studies.

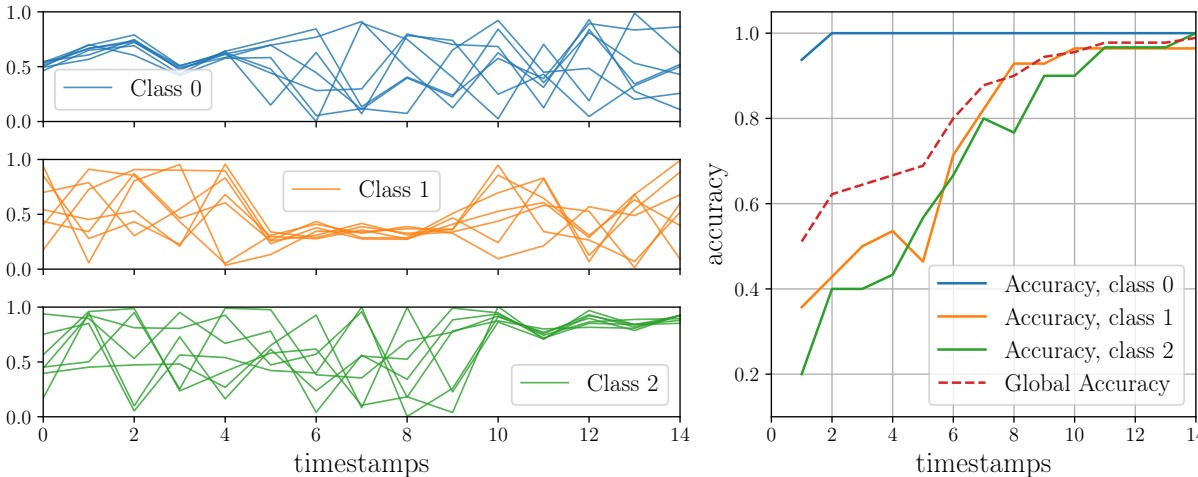

Figure 8: Examples of training time series of the *SmoothSubspace* dataset. Each of the class has a discriminative signal within one third of the serie (see Appendix D for a code snippet reproducing these results, based on the proposed library).

## C End-to-end ECTS

### C.1 State-of-the-art

Table 5: Table of published end-to-end methods for the ECTS problem.

| References | Architecture | RL | DL | Cost-genericity |
|---|---|---|---|---|
| EDSC (Xing et al., 2011) | Shapelet | | | |
| EARLIEST (Hartvigsen et al., 2019) | LSTM | ✓ | ✓ | |
| DDQN (Martinez et al., 2020) | MLP | ✓ | ✓ | |
| DETSCNet (Chen et al., 2022) | TCN | | ✓ | |
| Benefitter (Shekhar et al., 2023) | LSTM | ✓ | ✓ | ✓ |
| CIS (Cao et al., 2023) | LSTM | ✓ | ✓ | ✓ |
| ELECTS (Rußwurm et al., 2023) | LSTM | | ✓ | |
| EarlyStop-RL (Wang et al., 2024) | MLP | ✓ | ✓ | ✓ |

• A different class of methods relies on searching telltale representations of subsequences, such that if the incoming time sequence $\mathbf{x}_t$ matches one or more of these representations, then its class can be predicted. Typically, these representations take the form of shapelets that discriminate well one class from the others (Ye & Keogh, 2011). For instance, the Early Distinctive Shapelet Classification (EDSC) method learns a distance threshold for each shapelet, based on the computation of the Euclidean distance between the considered subsequence and all other valid subsequences in the training set (Xing et al., 2011). It selects a subset of them, based on a utility measure that combines precision and recall, weighted by the earliness. A prediction is made as soon as $\mathbf{x}_t$ matches one of these shapelets well enough. Because this family of methods is computationally expensive, extensions have been developed

to reduce the computational load (Yan et al., 2020; Zhang & Wan, 2022). Other extensions aim at improving the reliability of the predictions (Ghalwash et al., 2014; Yao et al., 2019), and tackling multivariate time series (Ghalwash & Obradovic, 2012; He et al., 2013; 2015; Lin et al., 2015).

• The EARLIEST (Early and Adaptive Recurrent Label ESTimator) uses a RNN architecture (Hartvigsen et al., 2019) to make the prediction and a Reinforcement Learning agent trained jointly using policy gradient to trigger prediction or not. If a prediction is triggered, the hidden representation given by the RNN is sent to a Discriminator, whose role is to predict a class, given this representation. The model has been adapted to deal with irregularly sampled time series (Hartvigsen et al., 2022).

• Martinez et al. (2018; 2020) use a Deep Q-Network (Mnih et al., 2015), alongside a specifically designed reward signal, encouraging the agent to find a good trade-off between earliness and accuracy. Those types of approaches also naturally extend to online settings where time series are not of fixed length.

• The Decouple ETSC Network (DETSCNET) (Chen et al., 2022) architecture leverages a gradient projection technique in order to jointly learn two sub-modules: one for variable-length series classification, and the other for the early exiting task.

• The BENEFITTER algorithm (Shekhar et al., 2023) is an anticipation-based approach that learns to predict the *benefit* of triggering a prediction early. This quantity is equal to the saving one could make by triggering some decision now minus the cost induced by a wrong prediction. A LSTM model is learned to regress the benefit, which thus triggers, at inference time, as soon as the benefit is positive, i.e. when savings induced by temporal costs exceed estimated misclassification costs.

• The Classifier-Induced Stopping (CIS) (Cao et al., 2023) model leverages policy gradient Reinforcement Learning in order to directly predict the optimal stopping time in a supervised way, found *a posteriori* for the training time series.

• The End-to-end Learned Early Classification of Time Series method (ELECTS) leverages an LSTM architecture, adding a stopping prediction head to the network and adapting the loss function to promote good early predictions (Rußwurm et al., 2023).

• Wang et al. (2024) introduce EARLYSTOP-RL, in which model-free RL is used to address the problem of early diagnosis of lung cancer.

## C.2 End-to-end experiments

In this section, we test out two popular ECTS end-to-end methods, i.e. EARLIEST (Hartvigsen et al., 2019) and ELECTS (Rußwurm et al., 2023), and directly compare them to the separable baseline PROBA THRESHOLD. Due to the fact that those method do not directly expand to generic cost settings, we only perform this experiment using the standard cost setting, as described in Section 4.2.

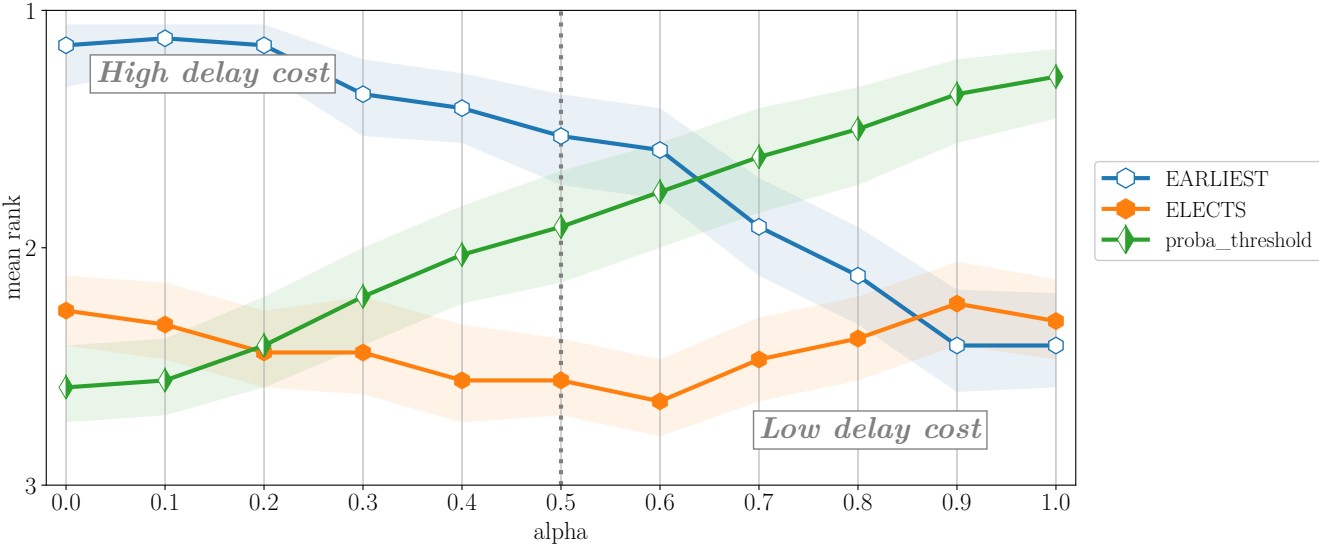

Figure 9: Standard cost setting, non z-normalized proposed datasets (orange cylinder). It is found that end-to-end approaches outperform the separable baseline PROBA THRESHOLD for high temporal costs. Since end-to-end approaches are not constrained to consider time series only over the 20 timestamps corresponding to every 5% of the time series duration for prediction, they can make their prediction in the interval. For instance, before 5% of the time series has been observed, which happens for high delay cost, whereas separable methods, in this experimental setting, must wait until the first time step, resulting in higher average costs. However, as the delay cost decreases, end-to-end methods tend to be less stable overall, falling behind simple separable base method, such as PROBA THRESHOLD.

## D Library

More details are given about the proposed library here. It leverages the modularity of separable ECTS approaches and offers the possibility to easily assess each of the component contribution to final performance. The main library object is an `EarlyClassifier` object that needs, at least, 3 inputs:

- a `ChronologicalClassifier` that outputs classification prediction based on varying-size time series,

- a `TriggerModel` that decides to whether accept or not the current classifier's prediction,

- a `CostMatrices` that defines the cost functions to be optimized.

As used throughout the paper, one way to implement a classification strategy is to use an ensemble of classifiers, each of which is specialized for a particular timestamp. The `ClassifiersCollection` object offers an interface to do that. It takes as input any scikit-learn estimator as well as a set of timestamps for which to learn a classifier. For example, the following code uses the same classification module used to produce Figure 8.

```python
from sklearn.linear_model import RidgeClassifierCV
from sklearn.preprocessing import SplineTransformer
from sklearn.calibration import CalibratedClassifierCV
from sklearn.pipeline import make_pipeline

from ml_edm.classification.classifiers_collection import ClassifiersCollection

T # time series' length
valid_timestamps = list(range(1, T+1))

clf = make_pipeline(
    SplineTransformer(),
```

```
    CalibratedClassifierCV(RidgeClassifierCV(), method="sigmoid")
)
collection_clf = ClassifiersCollection(
    base_classifier=clf,
    timestamps=valid_timestamps)
)
```

Then, having defined a classification strategy, one can fit a full ECTS model, for example, using the `ProbabilityThreshold` trigger model. By default, when no other argument than timestamp is given to the `CostMatrices` object, it uses the default cost setting, as defined in Section 4.2.

```
from ml_edm.early_classifier import EarlyClassifier
from ml_edm.trigger import ProbabilityThreshold
from ml_edm.cost_matrices import CostMatrices

early_clf = EarlyClassifier(
    chronological_classifiers=collection_clf,
    trigger_model=ProbabilityThreshold(valid_timestamps),
    cost_matrices=CostMatrices(valid_timestamps),
)
early_clf.fit(X, y)
```

Thus, the library allows to easily pursue research in the separable ECTS field, facilitating the implementation of new approaches (whether in the classification or trigger part) and the systematic comparison with previous ones. It also pave the way for end-to-end approaches that can be framed as a trigger model that does not rely on any classifier's outputs.

# E   Data description

## E.1   UCR Time Series Classification datasets

Table 6: UCR TSC datasets : 77 datasets from the UCR archive have been retained to run the experiments over the 128 contained in the full archive. Those are the ones with fixed length, without missing values and with enough training samples to execute our experiments pipeline end-to-end. *Italic* datasets are not included in experiments using default split for this reason.

| Data | Train | Test | Length | Class | Type |
|---|---|---|---|---|---|
| ACSF1 | 100 | 100 | 1460 | 10 | Device |
| Adiac | 390 | 391 | 176 | 37 | Image |
| *Beef* | 30 | 30 | 470 | 5 | Spectro |
| BeetleFly | 20 | 20 | 512 | 2 | Image |
| BME | 30 | 150 | 128 | 3 | Simulated |
| Car | 60 | 60 | 577 | 4 | Sensor |
| CBF | 30 | 900 | 128 | 3 | Simulated |
| Chinatown | 20 | 345 | 24 | 2 | Traffic |
| ChlorineConcentration | 467 | 3840 | 166 | 3 | Sensor |
| CinCECGTorso | 40 | 1380 | 1639 | 4 | Sensor |
| Coffee | 28 | 28 | 286 | 2 | Spectro |
| Computers | 250 | 250 | 720 | 2 | Device |
| CricketX | 390 | 390 | 300 | 12 | Motion |
| CricketY | 390 | 390 | 300 | 12 | Motion |
| CricketZ | 390 | 390 | 300 | 12 | Motion |
| Crop | 7200 | 16800 | 46 | 24 | Image |

| | | | | | |
|---|---|---|---|---|---|
| *DiatomSizeReduction* | 16 | 306 | 345 | 4 | Image |
| DistalPhalanxOutlineCorrect | 600 | 276 | 80 | 2 | Image |
| Earthquakes | 322 | 139 | 512 | 2 | Sensor |
| ECG200 | 100 | 100 | 96 | 2 | ECG |
| *ECG5000* | 500 | 4500 | 140 | 5 | ECG |
| ECGFiveDays | 23 | 861 | 136 | 2 | ECG |
| ElectricDevices | 8926 | 7711 | 96 | 7 | Device |
| EOGVerticalSignal | 362 | 362 | 1250 | 12 | EOG |
| EthanolLevel | 504 | 500 | 1751 | 4 | Spectro |
| FaceAll | 560 | 1690 | 131 | 14 | Image |
| FaceFour | 24 | 88 | 350 | 4 | Image |
| FacesUCR | 200 | 2050 | 131 | 14 | Image |
| *FiftyWords* | 450 | 455 | 270 | 50 | Image |
| Fish | 175 | 175 | 463 | 7 | Image |
| FordA | 3601 | 1320 | 500 | 2 | Sensor |
| FreezerRegularTrain | 150 | 2850 | 301 | 2 | Sensor |
| GunPoint | 50 | 150 | 150 | 2 | Motion |
| Ham | 109 | 105 | 431 | 2 | Spectro |
| HandOutlines | 1000 | 370 | 2709 | 2 | Image |
| Haptics | 155 | 308 | 1092 | 5 | Motion |
| Herring | 64 | 64 | 512 | 2 | Image |
| HouseTwenty | 34 | 101 | 3000 | 2 | Device |
| InlineSkate | 100 | 550 | 1882 | 7 | Motion |
| InsectEPGRegularTrain | 62 | 249 | 601 | 3 | EPG |
| InsectWingbeatSound | 220 | 1980 | 256 | 11 | Sensor |
| ItalyPowerDemand | 67 | 1029 | 24 | 2 | Sensor |
| LargeKitchenAppliances | 375 | 375 | 720 | 3 | Device |
| Lightning2 | 60 | 61 | 637 | 2 | Sensor |
| Lightning7 | 70 | 73 | 319 | 7 | Sensor |
| *Mallat* | 55 | 2345 | 1024 | 8 | Simulated |
| Meat | 60 | 60 | 448 | 3 | Spectro |
| MedicalImages | 381 | 760 | 99 | 10 | Image |
| MelbournePedestrian | 1200 | 2450 | 24 | 10 | Traffic |
| MixedShapesRegularTrain | 500 | 2425 | 1024 | 5 | Image |
| MoteStrain | 20 | 1252 | 84 | 2 | Sensor |
| NonInvasiveFetalECGThorax1 | 1800 | 1965 | 750 | 42 | ECG |
| NonInvasiveFetalECGThorax2 | 1800 | 1965 | 750 | 42 | ECG |
| OSULeaf | 200 | 242 | 427 | 6 | Image |
| OliveOil | 30 | 30 | 570 | 4 | Spectro |
| PhalangesOutlinesCorrect | 1800 | 858 | 80 | 2 | Image |
| Plane | 105 | 105 | 144 | 7 | Sensor |
| PowerCons | 180 | 180 | 144 | 2 | Power |
| ProximalPhalanxOutlineCorrect | 600 | 291 | 80 | 2 | Image |
| RefrigerationDevices | 375 | 375 | 720 | 3 | Device |
| Rock | 20 | 50 | 2844 | 4 | Spectrum |
| ScreenType | 375 | 375 | 720 | 3 | Device |
| SemgHandGenderCh2 | 300 | 600 | 1500 | 2 | Spectrum |
| *ShapesAll* | 600 | 600 | 512 | 60 | Image |
| SmoothSubspace | 150 | 150 | 15 | 3 | Simulated |
| SonyAIBORobotSurface1 | 20 | 601 | 70 | 2 | Sensor |
| SonyAIBORobotSurface2 | 27 | 953 | 65 | 2 | Sensor |
| StarLightCurves | 1000 | 8236 | 1024 | 3 | Sensor |
| Strawberry | 613 | 370 | 235 | 2 | Spectro |

| | | | | | |
|---|---|---|---|---|---|
| SwedishLeaf | 500 | 625 | 128 | 15 | Image |
| *Symbols* | 25 | 995 | 398 | 6 | Image |
| SyntheticControl | 300 | 300 | 60 | 6 | Simulated |
| ToeSegmentation1 | 40 | 228 | 277 | 2 | Motion |
| Trace | 100 | 100 | 275 | 4 | Sensor |
| TwoLeadECG | 23 | 1139 | 82 | 2 | ECG |
| TwoPatterns | 1000 | 4000 | 128 | 4 | Simulated |
| UMD | 36 | 144 | 150 | 3 | Simulated |
| UWaveGestureLibraryX | 896 | 3582 | 315 | 8 | Motion |
| UWaveGestureLibraryY | 896 | 3582 | 315 | 8 | Motion |
| UWaveGestureLibraryZ | 896 | 3582 | 315 | 8 | Motion |
| Wafer | 1000 | 6164 | 152 | 2 | Sensor |
| Wine | 57 | 54 | 234 | 2 | Spectro |
| *WordSynonyms* | 267 | 638 | 270 | 25 | Image |
| Worms | 181 | 77 | 900 | 5 | Motion |
| Yoga | 300 | 3000 | 426 | 2 | Image |

## E.2 Proposed, non z-normalized, datasets

Table 7: New datasets collection: 34 datasets from both the UCR archive (dashed line) and the Monash UEA extrinsic regression archive. When missing values and/or varying lengths, replace missing values with 0 and pad series to maximum length with 0. All of the datasets are not *z*-normalized. AUC gain is mean improvement, aggregated over a time step window of length 5, e.g. in the first line, mean test AUC gets 11% better when using $[40\%, ..., 60\%]$ and 16% better with $[75\%, ..., 100\%]$ compared to $[5\%, ..., 25\%]$ of the series. *Italic* datasets are not included when classes are imbalanced as problems become too difficult for the chosen classifiers.

| Data | Size | Length | Class | Type | AUC Gain train (half/full) | AUC Gain test (half/full) |
|---|---|---|---|---|---|---|
| BME | 180 | 128 | 3 | Simulated | (7%/7%) | (11%/16%) |
| Chinatown | 365 | 24 | 2 | Traffic | (1%/1%) | (0%/0%) |
| Crop | 24000 | 46 | 24 | Image | (9%/10%) | (8%/9%) |
| DodgerLoopDay | 158 | 288 | 7 | Sensor | (4%/5%) | (14%/17%) |
| EOGVerticalSignal | 724 | 1250 | 12 | EOG | (35%/35%) | (43%/45%) |
| GestureMidAirD1 | 338 | 360 | 26 | Trajectory | (11%/12%) | (23%/26%) |
| GunPointAgeSpan | 451 | 150 | 2 | Motion | (7%/7%) | (7%/7%) |
| HouseTwenty | 135 | 3000 | 2 | Device | (1%/1%) | (5%/6%) |
| MelbournePedestrian | 3650 | 24 | 10 | Traffic | (15%/15%) | (15%/16%) |
| PLAID | 1074 | Vary | 11 | Device | (0%/0%) | (2%/2%) |
| Rock | 70 | 2844 | 4 | Spectrum | (1%/1%) | (12%/12%) |
| SemgHandGenderCh2 | 900 | 1500 | 2 | Spectrum | (1%/2%) | (11%/13%) |
| SmoothSubspace | 300 | 15 | 3 | Simulated | (21%/37%) | (20%/38%) |
| UMD | 180 | 150 | 3 | Simulated | (10%/11%) | (22%/28%) |
| AcousticContaminationMadrid | 138 | 365 | 2 | Environment | (1%/2%) | (5%/7%) |
| AluminiumConcentration | 629 | 2542 | 2 | Environment | (3%/4%) | (5%/10%) |
| BitcoinSentiment | 332 | 24 | 2 | Sentiment | (11%/13%) | (-5%/-6%) |
| ChilledWaterPredictor | 459 | 168 | 2 | Energy | (0%/0%) | (8%/11%) |
| *CopperConcentration* | 629 | 2542 | 2 | Environment | (4%/5%) | (4%/3%) |
| *Covid19Andalusia* | 204 | 91 | 2 | Health | (7%/8%) | (7%/17%) |
| *DailyOilGasPrices* | 188 | 30 | 2 | Economy | (25%/23%) | (10%/8%) |
| DhakaHourlyAirQuality | 2068 | 24 | 2 | Environment | (2%/3%) | (0%/2%) |
| ElectricityPredictor | 810 | 168 | 2 | Energy | (4%/5%) | (7%/13%) |
| FloodModeling3 | 613 | 266 | 2 | Environment | (20%/22%) | (20%/31%) |
| HouseholdPowerConsumption1 | 1431 | 1440 | 2 | Energy | (4%/7%) | (14%/27%) |
| HotwaterPredictor | 351 | 168 | 2 | Energy | (1%/1%) | (4%/5%) |
| MadridPM10Quality | 6923 | 168 | 2 | Environment | (4%/5%) | (11%/16%) |
| ParkingBirmingham | 1888 | 14 | 2 | Environment | (7%/28%) | (5%/22%) |
| PrecipitationAndalusia | 672 | 365 | 2 | Environment | (0%/1%) | (3%/4%) |
| *SierraNevadaMountainsSnow* | 500 | 30 | 2 | Environment | (6%/7%) | (-2%/4%) |
| SolarRadiationAndalusia | 672 | 365 | 2 | Energy | (1%/1%) | (4%/4%) |
| SteamPredictor | 300 | 168 | 2 | Energy | (2%/2%) | (3%/6%) |
| TetuanEnergyConsumption | 364 | 144 | 2 | Energy | (5%/5%) | (1%/6%) |
| WindTurbinePower | 852 | 144 | 2 | Energy | (11%/12%) | (15%/21%) |

## F  Supplementary results

### F.1  Additional figures : Standard cost setting

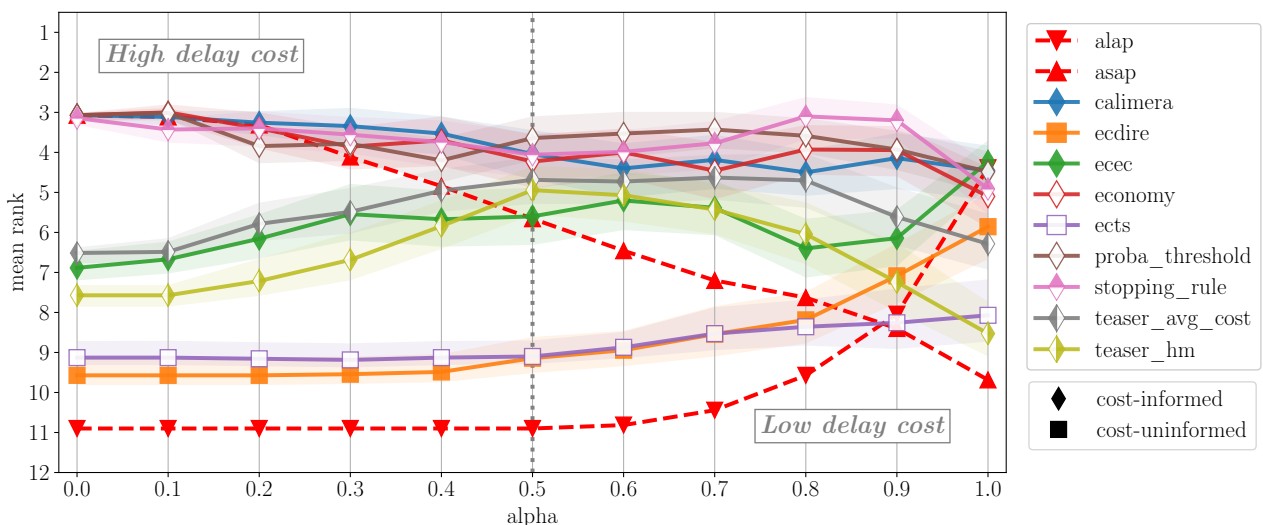

Figure 10: Standard cost setting, non z-normalized proposed datasets (orange cylinder). Compared to Figure 3a, the global ranking is not altered much. One can observe that for $\alpha \in [0.5, 0.7]$ the top group is now more populated, gathering the first six approaches, probably due to the limited amount of datasets available in this case.

### F.2  Additional figures : Anomaly detection cost setting

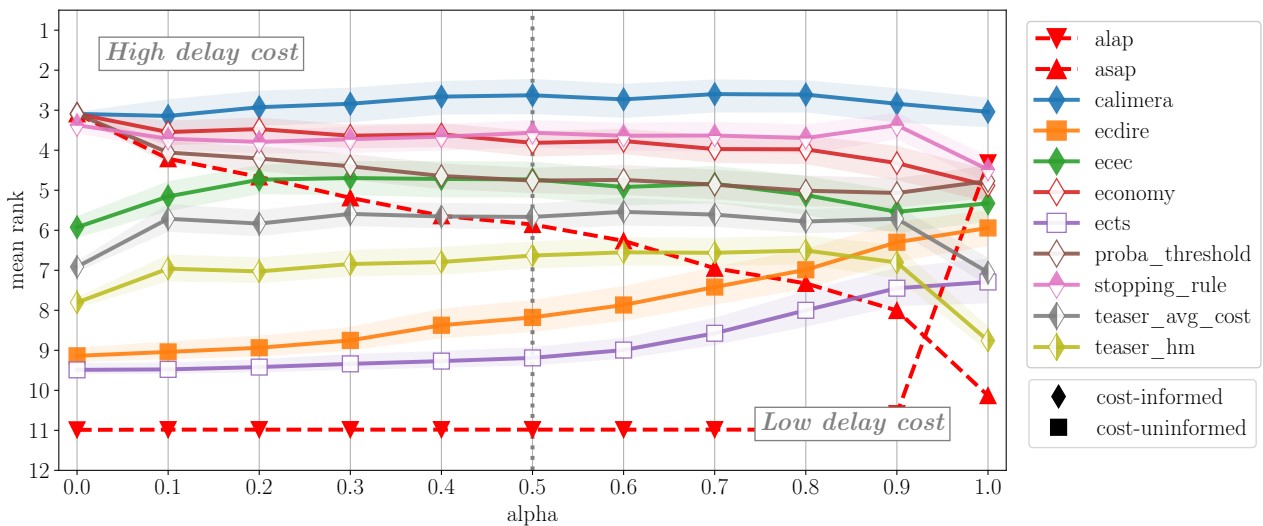

Figure 11: Anomaly detection cost setting, original UCR datasets (blue cylinder). Compared to Figure 6a, one can see that CALIMERA is now clearly dominating all other methods for all $\alpha$. The global ranking remains globally stable otherwise.

## F.3 Removing calibration

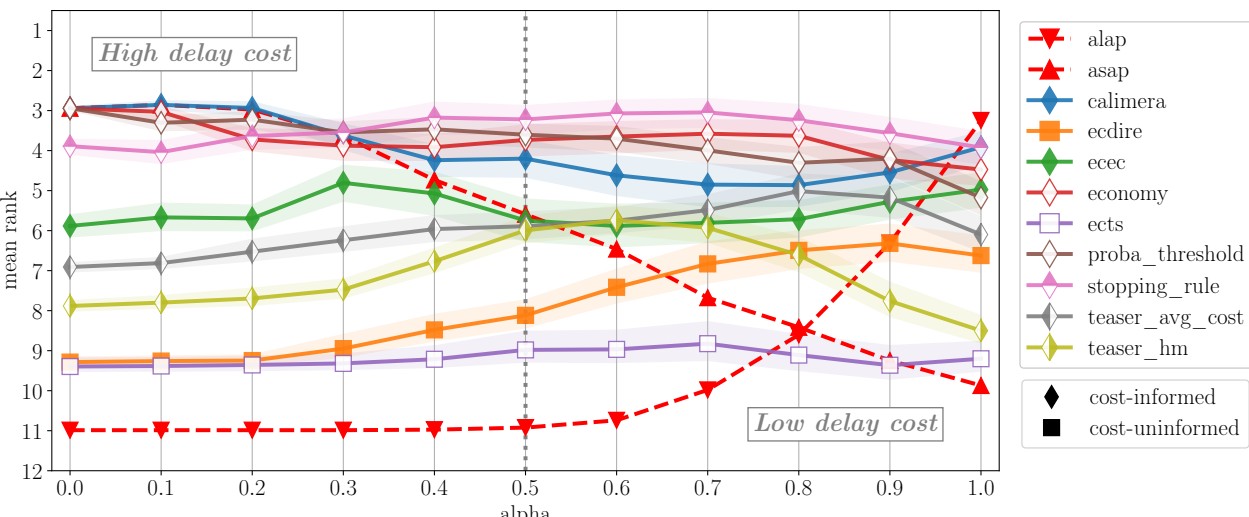

Figure 12: Standard cost setting, original UCR datasets (blue cylinder). The calibration step is now removed, i.e. the outputs from the decision function is now simply passed through a *softmax* function. Both CALIMERA and PROBA THRESHOLD suffer heavily from using uncalibrated scores.

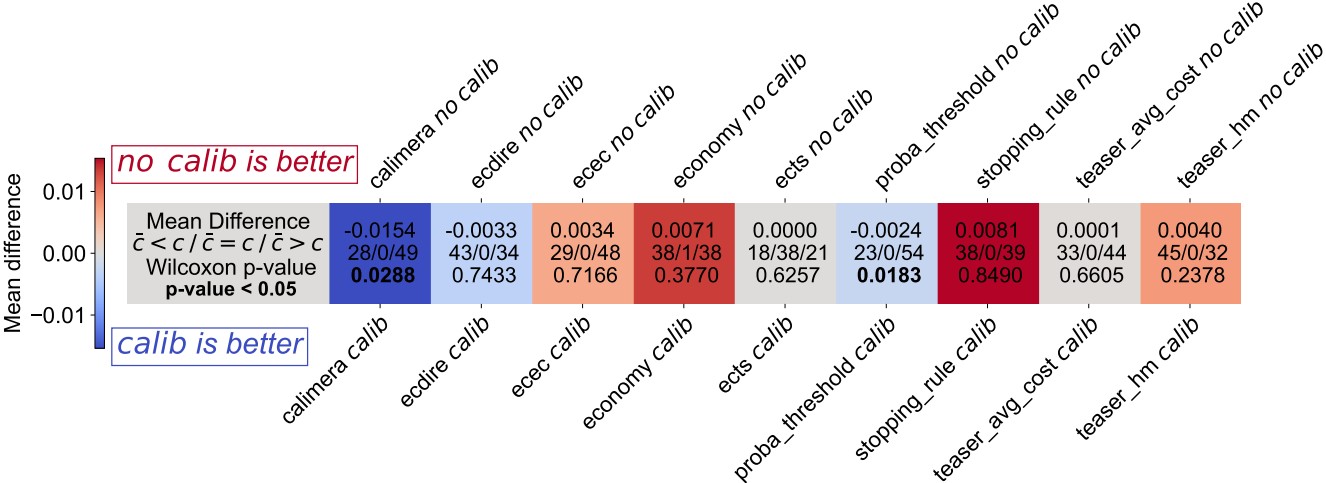

Figure 13: Pairwise comparison (Ismail-Fawaz et al., 2023), calibration (*calib / C*) vs no calibration (*no calib / $\bar{C}$*). We select $\alpha = 0.8$ as the alpha value where both the naive baselines cross, i.e. where, in average, most of datasets are more challenging. Square colors are indexed on the mean *AvgCost* difference. For example, CALIMERA has a lower mean *AvgCost* when trained over calibrated scores: it appears in dark blue. The Wilcoxon p-value is equal to 0.0288, which is lower than significance level equal to 0.05. Thus, CALIMERA statistically under-performs when using uncalibrated scores. This is also the case for the PROBA THRESHOLD method.

## F.4 Ablation study on non-myopic trigger models

One of the key component when it comes to trigger functions is the *non-myopic* ability to anticipate the likely future. In this section, we conduct an ablation study over the *anticipation-based* property by making those type of approaches myopic. In particular, ECONOMY and CALIMERA will be of interest here. More precisely, the expected

cost horizon has been limit to 1 for ECONOMY. Concerning CALIMERA, the backward cost propagation has been blocked such that the only available information to trigger be the expected cost difference between current and next timestamp.

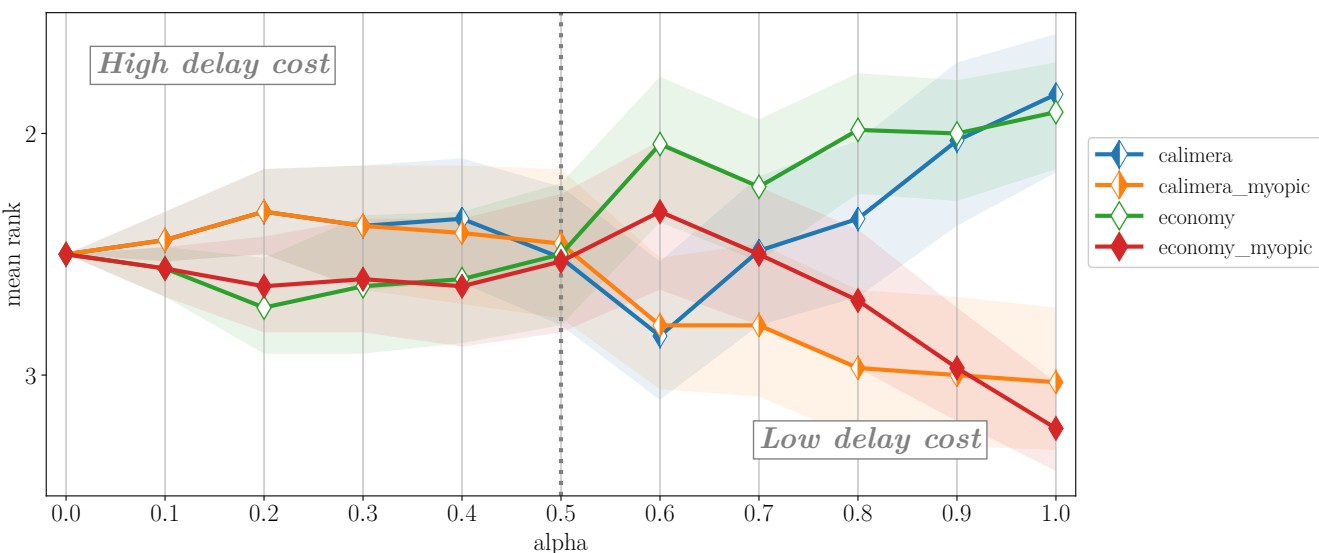

Figure 14: Standard cost setting, non z-normalized proposed datasets (orange cylinder). The *anticipation-based* property is blocked from non-myopic methods. The myopic counterparts perform equal when dealing with high temporal cost, but are significantly worse when it gets weaker, i.e. for $\alpha > 0.5$.

### F.5 Changing the base classifier

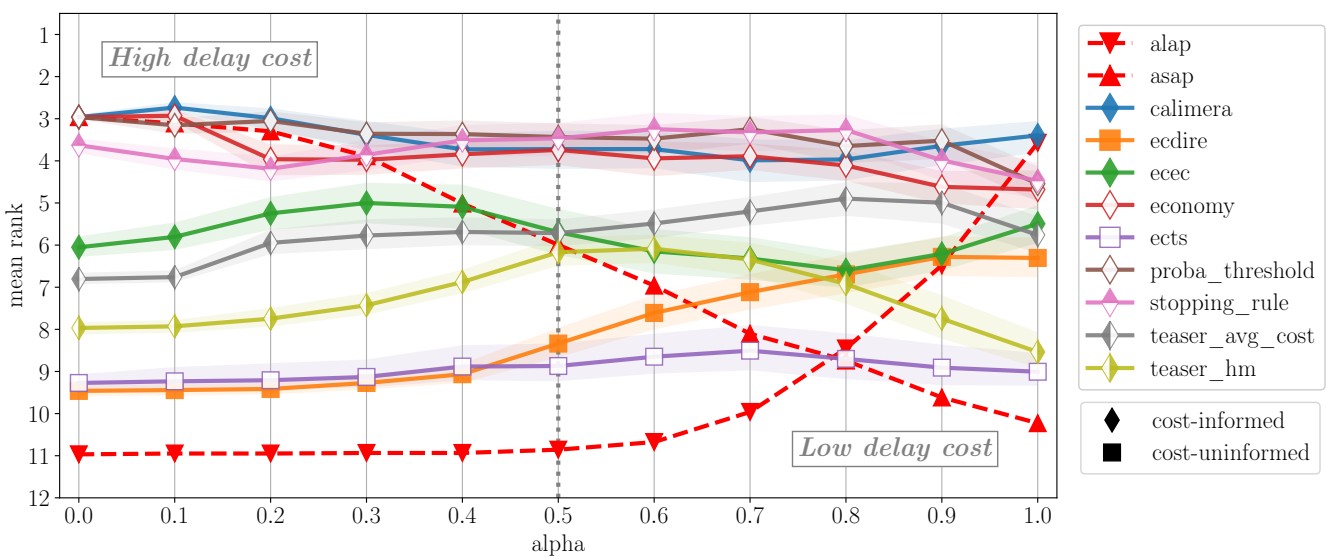

Figure 15: Standard cost setting, original UCR datasets (blue cylinder). The base classifier is now WEASEL 2.0 Schäfer & Leser (2023). Results are very close to those exposed in Figure 3a.

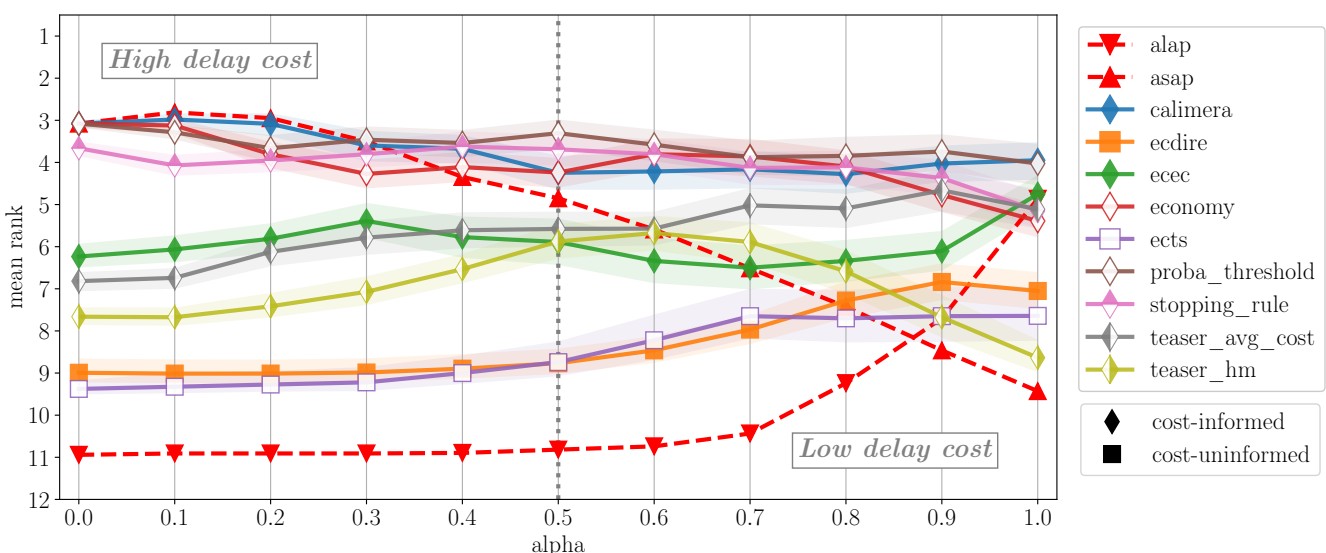

Figure 16: Standard cost setting, original UCR datasets (blue cylinder). The base classifier is now a pipeline including features extraction with TSFRESH (Christ et al., 2018) and classification using XGBoost (Chen et al., 2015). Results are a bit noisier than those exposed in Figure 3a.

Table 8: Comparison of the tested classifiers. Percentage representing, for each alpha, the amount of dataset for which each classifier is ranked first, averaged over all trigger models and all datasets. Ties are not considered ; thus, each line may sum to less than 1. Best performing classifier is underlined.

| | classifier | | |
|---|---|---|---|
| $\alpha$ | MINIROCKET | WEASEL 2.0 | TSFRESH&XGBOOST |
| 0 | 12.60% | 11.95% | 16.36% |
| 0.1 | 31.56% | 18.44% | 37.01% |
| 0.2 | 32.86% | 18.96% | 36.49% |
| 0.3 | 36.49% | 19.48% | 32.34% |
| 0.4 | 39.87% | 17.53% | 31.30% |
| 0.5 | 43.12% | 19.10% | 26.62% |
| 0.6 | 42.60% | 23.25% | 23.51% |
| 0.7 | 44.16% | 24.94% | 20.52% |
| 0.8 | 48.44% | 25.45% | 15.71% |
| 0.9 | 48.70% | 26.62% | 14.29% |
| 1 | 42.21% | 26.75% | 14.68% |

## F.6 Impact of z-normalization

Clearly, using z-normalized datasets is not applicable in practice, as it would require knowledge of the entire incoming time series. In a research context, previous work has used such training sets to test the proposed algorithms. Our goal here, is to assess whether this could have a large impact on the performances. For example, when a normalized time series has a low variance at the beginning, we can expect a high variance in the rest of the series since the mean variance is 1. There is therefore an information leakage that can be exploited by an ECTS algorithm, while this is not representative of what happens in real applications. A proposal such as the one presented by Schäfer & Leser (2020), where the z-normalization of available time series is repeated at each time step, has its own problems. In particular, it means that if a single classifier is used for all time steps, the representation of $\mathbf{x}_t$ can be different at times $t$ and $t+1$ and all further time steps which can induce confusion for the classifier.

On the one hand, z-normalization induces an information leakage that could help methods to unduly exploit knowledge about the future of incoming time series. On the other hand, any normalization rescales the signal and therefore, potentially, hinder the recognition of telltale features. So, does z-normalization affect the performance of ECTS methods? And if yes, in which way?

In order to answer this question, we took the new datasets collection described in Section 4.1. They are indeed not z-normalized originally. We duplicate and z-normalized them to get a second collection. As explained in Section 4.1, some extrinsic regression datasets have been converted into classification ones. Here, the threshold value chosen to discretize the output into binary classes has been set to the median of the regression target. In this way, classes within those datasets are equally populated.

In these experiments, the delay cost is linear as in Section 4.2 and as in most of the literature. Figure 17 reports pairwise comparisons done on the 35 datasets. We look at $\alpha = 0.8$, as this is the only value for which significant differences are observed. One can see that most of the trigger models do not actually benefit from the z-normalization. Quite the opposite: out of nine trigger models, only one, i.e. ECDIRE, actually has a better mean $AvgCost$ when being trained on z-normalized data. Regarding the remaining methods, both STOPPING RULE and TEASER$_{Avg}$ perform significantly worse when operating on z-normalized data. Those trends are quite similar for other $\alpha$ values, without any significance on the statistical tests though. Thus, while z-normalization has some impact, since privileged information from the future can be leaked, our experiments, for the proposed datasets collection at least, show that this does not alter the overall results reported in the literature, and are globally in accordance with the results presented in Section 4.

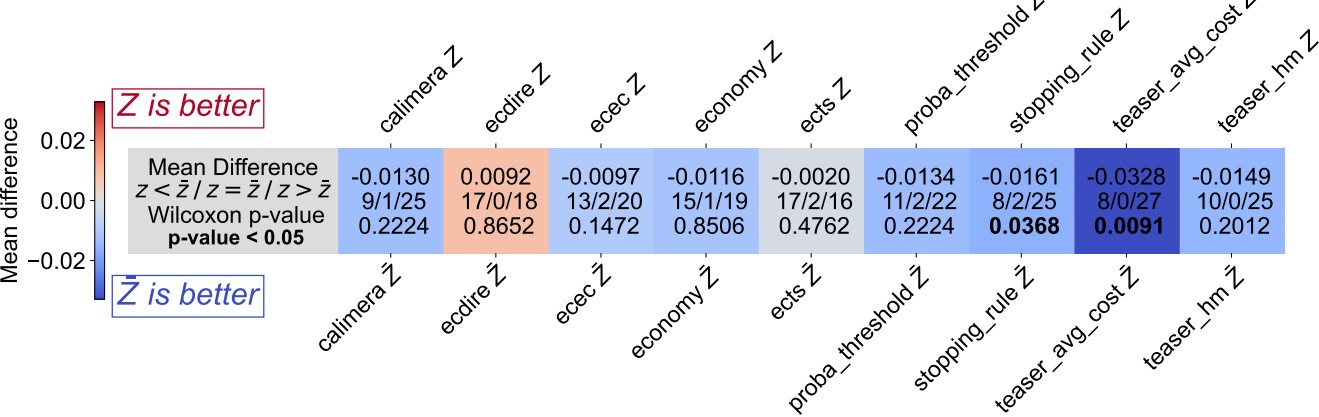

Figure 17: Pairwise comparison (Ismail-Fawaz et al., 2023), $z$-normalization ($Z$) vs no $z$-normalization ($\bar{Z}$), $\alpha = 0.8$. Square colors are indexed on the mean $AvgCost$ difference. For example, CALIMERA has a lower mean $AvgCost$ when trained over non $z$-normalized datasets by 1.3e-2 and appears in light blue. It beats the $z$-normalized version over 25 datasets, loses over 9 and are tied on 1. The Wilcoxon p-value is equal to 0.2224, which is higher than significance level equal to 0.05. Thus, no statistical difference can be observed for the considered approach.

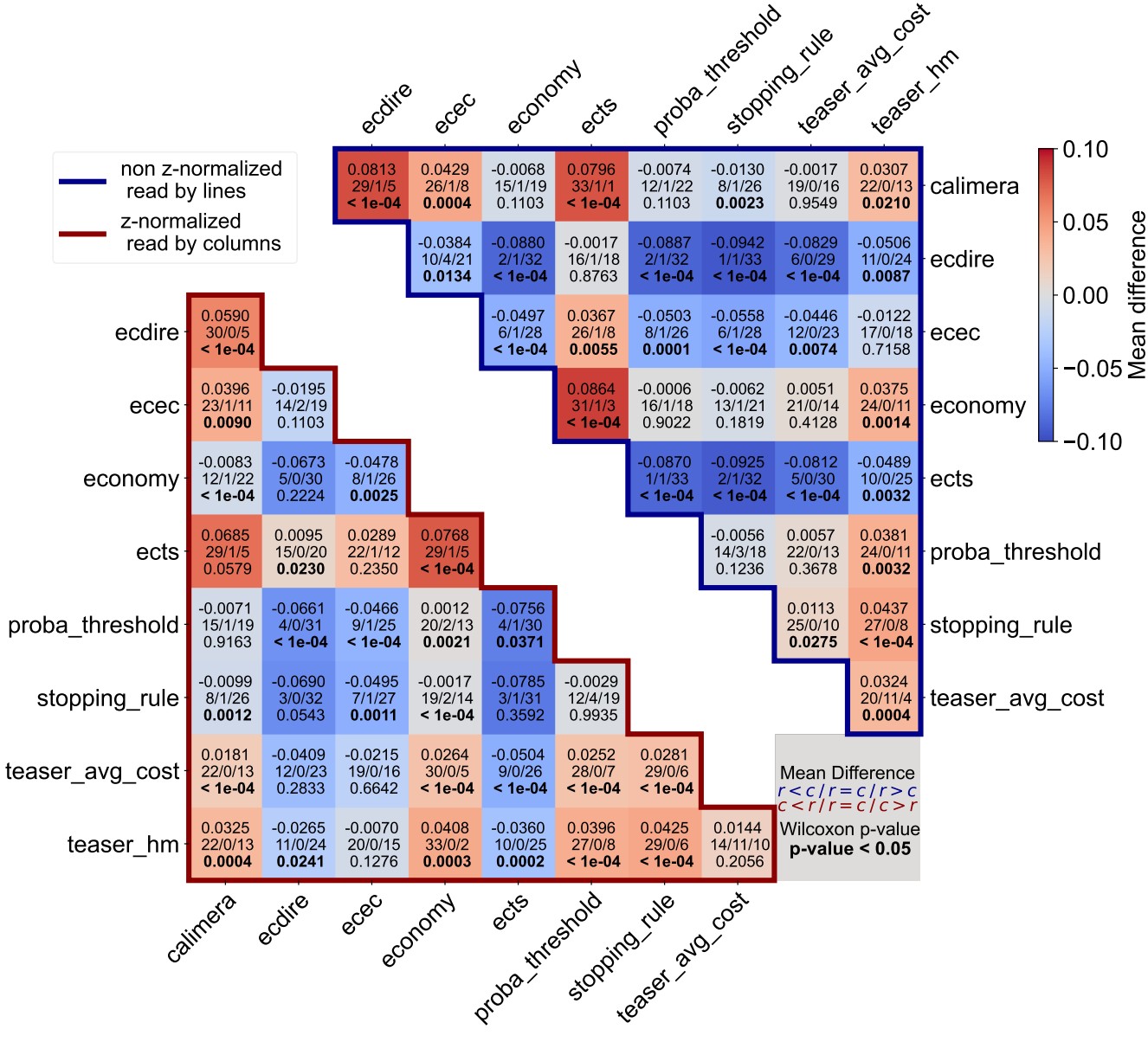

Figure 18: Multi-comparison-matrices (Ismail-Fawaz et al., 2023). The upper triangle, with dark blue contours, displays the comparison of the competitive methods trained over non z-normalized dataset (orange cylinder). The values within this triangle has to be read *by lines*, i.e. for a considered line, red shades indicate better performances, blue shades weaker performances. The lower triangle, with dark red contours, is the comparison of the methods trained over the same datasets, z-normalized (chocolate cylinder). The values within this triangle has to be read *by columns*, i.e. for a considered column, red shades indicate better performances, blue shades weaker performances. The complete figure being symmetrical indicates that z-normalization does not impact much relative ranking between methods.

### F.7 Anomaly detection cost setting : an ablation study

**Exponential delay cost only**

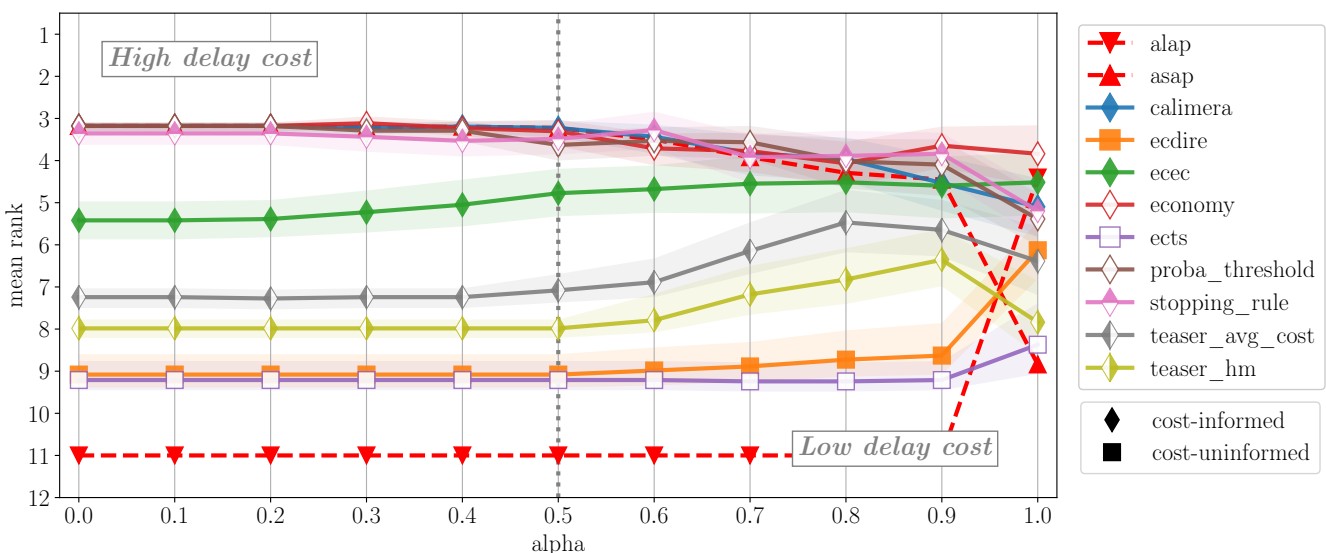

Figure 19: Exponential delay cost, symmetric binary misclassification cost, non z-normalized proposed imbalanced datasets (orange cylinder with a whole).

**Imbalanced misclassification cost only**

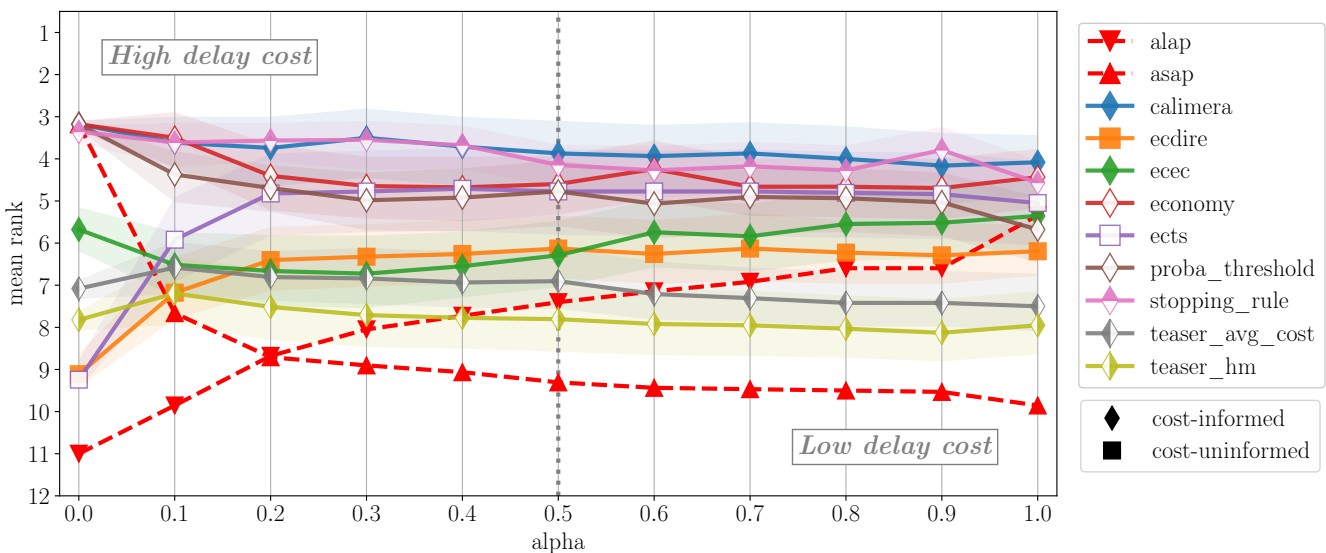

Figure 20: Linear delay cost, non symmetric imbalanced misclassification cost, non z-normalized proposed imbalanced datasets (orange cylinder with a whole).

**Standard cost setting, imbalanced datasets**

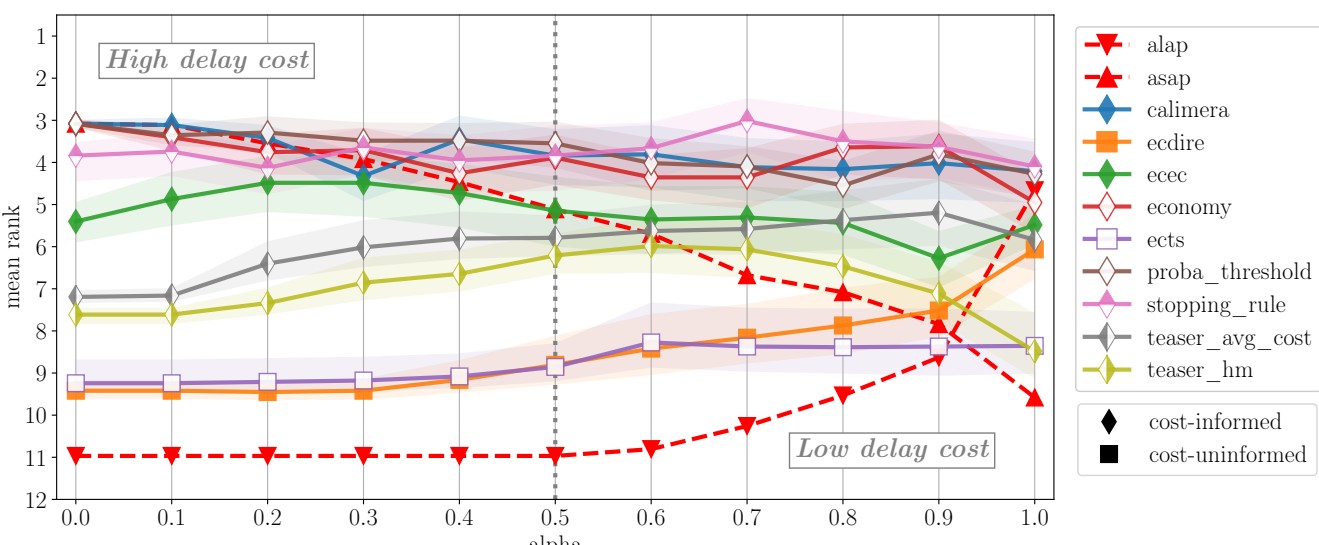

Figure 21: Linear delay cost, symmetric binary misclassification cost, non z-normalized proposed imbalanced datasets (orange cylinder with a whole).

