# OpenReview forum: "Early Classification of Time Series: A Survey and Benchmark"
_TMLR — Accepted by TMLR_

### Review · Reviewer_evND · 2025-07-02

**Summary Of Contributions:**

This paper discusses the problem, "early classification of time series (ECTS)", to construct a model that predicts a class label as early as possible with a smaller misclassification cost, based on repeated sequential observations of a certain phenomenon. The authors are concerned about a lack of a systematic and shared evaluation protocol to compare the relative merits of various methods.

Therefore, the authors first listed and organized general components of ECTS methods, to clarify the differences between various methods, as follows:
ECTS methods can be end-to-end or separable, and separable ECTS methods consist of a trigger function and a classification function.
A trigger function is characterized by either "cost-informed" ("at training" or "at test") or not ("cost-uninformed"), and either "anticipation-based", "myopic", or "blind" that is determined independently of learning, and a cost-uninformed trigger function is categorized into either "confidence-based" (either "instance-based" or "sequence-based") or "blind".
A classification function is characterized by whether it is "at each timestamp (classifier collection)" or "projection-based to treat all timestamps uniformly (single classifier)", in addition to the choice of base classifier such as SVM or NN.

The authors then discussed the significance of each characteristic of the ECTS method based on extensive experimental comparisons of various ECTS methods using many datasets. The findings are summarized in Sections 4.2.2, 4.3.2, 4.4, and 5, and the program code used will also be published.

This paper is a survey, and the contributions of this paper are
- characterization of components of ECTS methods,
- extensive experimental comparisons

to encourage future systematic development.

**Audience:**

Yes

**Claims And Evidence:**

Yes

**Requested Changes:**

Please change regarding W1,2,3,6,7,9.
This is critical to securing my recommendation for acceptance.

Please make changes the authors think would be beneficial regarding W4,5,8.
However, I would like you to consider revising for W4 strongly.
These would simply strengthen the work in my view.

**Strengths And Weaknesses:**

Strengths

S1.
The proposed characterization of components of ECTS methods may be natural. The previous work (Gupta et al., 2020) categorized ECTS methods into prefix based, shapelet based, model based, and miscellaneous approaches. I feel that the proposed characterization is more useful than that of (Gupta et al., 2020).

S2.
Experimental comparisons are clearly purposeful, sufficiently large, and reliable.

___

Weaknesses

(I could not find serious weaknesses in the 1st review, and the followings are minor weaknesses.)

W1.
The authors define the training data as $(\\mathbf{x}\_T^i,y^i)\_{i\\in[0,M]}$, which is a gross simplification (Sec.1, Para.3, P.1). For example, the authors described "The number of measurements, and hence the input dimension, varies." (Sec.2.4, Para.1, P.8), but this assertion does not fit with the simple definition of the training data. It's not that the assertion is hard to understand, but the writing style is a bit sloppy. They will need to either clearly generalize $T$ to $T\_i\in[0,T]$ or describe some kind of excuse.

W2.
There is a discrepancy in the notation for $\\mathcal{L}$: $\\mathcal{L}(\\cdot,y,t)$ above Equation (1) and $\\mathcal{L}(\\cdot,t,y)$ in Eq.(1).

W3.
Eq.(3) needs to include the randomness of $y$.

W4.
Figure 1 and Table 1 should be written in more detail. For example, they should include all the keywords used to characterize the trigger function such as "blind" or not, "confidence-based" ("instance-based" or "sequence-based") or not, "cost-informed" ("at training" or "at test") or not ("cost-uninformed"), and "anticipation-based" or not ("myopic"). Looking at Table 1, it is not clear whether "Confidence ✓" is "instance-based" or "sequence-based." Also, in Figure 1, can you use a tree structure to characterize the trigger function?

W5.
The authors write only "AvgCost is the most appropriate criterion to both train and evaluate ECTS approaches." under Eq.(4), while they write "We claim that the AvgCost is the appropriate measure by which to evaluate the performance of the methods. This is indeed what will be "paid" at the end of the day by a practitioner using a method." (Sec.5, Para.1, P.25). The authors should also describe a discussion of the appropriateness of AveCost under (4).

W6.
"which we call episodes" loses the comma (Sec.3.3, Para.2, P.13).

W7.
The authors should define "noisy cost" and "cost drift" or explain those with reference (Sec.5, P.26).

W8.
What methods do the authors envision for exploiting the misclassification cost in training? Is it something like the following?
- Pires, B. A., Szepesvari, C., & Ghavamzadeh, M. (2013, May). Cost-sensitive multiclass classification risk bounds. In International Conference on Machine Learning (pp. 1391-1399). PMLR.
- Charoenphakdee, N., Cui, Z., Zhang, Y., & Sugiyama, M. (2021, July). Classification with rejection based on cost-sensitive classification. In International Conference on Machine Learning (pp. 1507-1517). PMLR.

I think that such a topic is relevant to general classification methods and not specific to ECTS, so it may be redundant. If the authors believe that a discussion specific to ECTS is possible, authors should provide more details.

W9.
Formatting Instructions for TMLR Journal Submissions (Latex template) writes "The table number and title always appear before the table". Change the location of captions for Tables 1-5.

---

> ### Author Response · Authors · 2025-07-17
>
> We would like to thank the reviewer for his time and valuable feedback. Here are some clarifications and changes we made based on pointed weaknesses.
>
> **W1**: Mentioned sentence in (Sec2.4, Para.1) has been updated to clarify the fact that the input dimension varies at testing time, when progressively observing the time series. The training data, as defined in the paper, are indeed complete time series, whose length $T$ is the same for all examples. The "Problem statement" paragraph has been adapted to emphasize it.
>
> **W4**: We have updated Table 1 so as to include the terms "instant" and "sequence" as those terms indeed appeared in Fig 1. Though, for readability, we keep the absence of checkmark ($\checkmark$) to signify that methods don't have one or other property. For example, methods with no checkmarks in the *anticipation-based* column are thus myopic.
> Moreover, we choose not to use a tree structure to represent the trigger properties as we think that it may be ambiguous, as all of the mentioned properties are not mutually exclusive; as such, some trigger function could be confidence-based, anticipation-based and cost-informed at training & testing time all at once.
>
> **W5**: Some sentence has been added under Eq.(4)
>
> **W7**: Examples with more details have been added to the conclusion (Sec.5, Para.5) to describe both "noisy costs" and "cost drift".
>
> **W8**: We don't believe there's anything particular to discuss  in including decision costs to the classification module of ECTS methods.
>
> **W2**, **W3**, **W6**, **W9** have been dealt with in the newly revised version.
>
> Thank you again for your comments. If there're any further questions, let us know, we'll be glad to answer them.

---

> > ### Comment · Reviewer_evND · 2025-09-30
> > **2nd review comment**
> >
> > I get satisfied with the responses for W1-9, but I found a new weakness:
> >
> > W10. A table in p.18 of the revised paper loses a caption.

---

> > > ### Author Response · Authors · 2025-09-30
> > >
> > > Thank you for your comment, the caption of Table in p.18 has been added.

---

### Review · Reviewer_sR8Z · 2025-08-29

**Summary Of Contributions:**

This work makes an impactful contribution to the early classification of time series field from a perspective of survey paper.

(1) It proposes a principle-based taxonomy for early classification of time series methods that systematically organizes existing approaches by four key dimensions.

(2) It conducts extensive, rigorous experiments to benchmark 9 state-of-the-art early classification of time series algorithms across diverse scenarios, addressing past evaluation flaws.

(3) It develops an open-source library that implements most existing algorithms  and integrates the proposed evaluation protocol.

(4) It introduces a collection of 34 non-z-normalized datasets to address information leakage in traditional evaluations.

**Audience:**

Yes

**Claims And Evidence:**

Yes

**Requested Changes:**

*The paper should extend its experiments to include 3–5 representative end-to-end methods to provide a complete state-of-the-art comparison.

*To address the limited diversity and scale of non-z-normalized datasets, the paper is suggested to: (1) Add new datasets from high-stakes domains to increase domain coverage. (2) Include datasets with varied characteristics to test method robustness. (3) Validate converted regression datasets by consulting domain experts.

*To fill the gap in component-level analysis, the paper is suggested to perform ablation studies on the key components of 3–4 benchmarked trigger functions.

**Strengths And Weaknesses:**

**Strengths**

*The proposed taxonomy is not arbitrary but rooted in ECTS’s core challenges, ensuring it captures the essence of different methods. The evaluation protocol further enhances rigor by standardizing key variables.

*The paper prioritizes scenarios that matter for industry and healthcare. This practical focus ensures the paper’s findings can be directly applied to solve industrial problems.

*The open-source library and detailed experimental documentation set a new standard for ECTS research. The library includes full code. The paper also reports supplementary results in appendices, leaving no gaps in the evaluation.

**Weaknesses**

*The paper explicitly excludes end-to-end methods from experiments, stating they are “left for future work.” This is a major limitation, as end-to-end methods which integrate classification and triggering into a single model, often using deep learning have gained traction in recent years. By ignoring these methods, the benchmark fails to provide a complete picture of state-of-the-art ECTS performance. This omission also limits the paper’s utility for researchers working on deep learning-based ECTS, who cannot compare their methods to the benchmarked separable approaches.

*While the paper introduces 34 non-z-normalized datasets, this collection has two flaws. First, it is small compared to the 77 UCR datasets used in standard settings, reducing the statistical power of experiments on anomaly detection. Second, the datasets lack diversity in key characteristics: most are from energy or environment domains, with few from high-stakes fields like healthcare or finance. This limits the generalizability of the paper’s findings. Additionally, the paper converts some regression datasets to classification by thresholding, but it does not validate whether these converted datasets reflect real-world classification tasks.

*The paper tests two cost settings (balanced/linear and unbalanced/exponential) but ignores more realistic, complex cost functions common in practice. In healthcare, delay cost may depend on the true class, or misclassification cost may be stochastic. The paper only mentions these scenarios as “future work” but provides no analysis of how existing methods would adapt. This limitation reduces the paper’s relevance to real-world applications, where cost functions are rarely static or independent of class.

*While the paper conducts ablation studies on classifier calibration and base classifiers, it does not analyze the impact of individual components within trigger functions.

---

> ### Author Response · Authors · 2025-09-26
>
> We would like to thank the reviewer for his time and valuable feedback. Here are some clarifications and changes we made based on requested changes.
>
> **Requested changes**:
>
> **1** - The scope of the paper is about separable approaches, however we have tested two end-to-end approaches and the related results are reported in Appendix A.2.
>
> **2** - The reported comparison is made on 34 datasets, which is a significant increase over previous those carried out in previous survey papers (see the paragraph *Position with respect to other literature surveys* at the end of the introduction Section). We acknowledge that one important component of future work should be to offer to the community more data sets taken from a spectrum of application domains, which is now mention in the conclusion section.
>
> **3** - One of the key component of trigger models is non-myopia; we conduct an additional ablation study over the anticipation-based models Economy and Calimera (see Section 4.4), in order to assess the importance of this property. Detailed results are displayed in Appendix E.4.

---

> > ### Comment · Reviewer_sR8Z · 2025-09-30
> > **Thank you for your responses.**
> >
> > I have carefully read the authors' responses and have no further questions.

---

### Review · Reviewer_zkBX · 2025-09-15

**Summary Of Contributions:**

This paper discusses methods for early classification in time series (ECTS). It introduces several criteria by which one can classify ECTS methods. The paper distinguishes methods that consist of a separate classification and decision component, and methods that work end-to-end. The paper classifies separable approaches based on whether they use the cost during training, whether they use the cost during inference, how much context is used in the decision component, whether the model relies on predictions, and how many classifiers are used. The paper classifies 20 methods with these criteria. Furthermore, the paper curates a collection of datasets for benchmarking ECTS methods and presents two experiments where 11 methods are compared including a novel method introduced in this paper. Additional experiments are provided in the appendix.

**Audience:**

Yes

**Claims And Evidence:**

No

**Requested Changes:**

In my opinion the paper could be significantly improved by shortening it considerably (maybe 12 pages would be enough), removing repetition, and increasing precision and clarity. In my opinion, the paper should focus on the taxonomy and the evaluation and less on the survey. Section 3 could be moved to the appendix, where it could be extended to provide more details on each method, in particular, in the context of the proposed taxonomy. Most importantly, the paper should provide convincing evidence and citations for all claims, including a thorough discussion of related work. The evaluations could be improved with more methods and a more precise discussion of the setup of the evaluation protocol, in particular, regarding the selection of methods and evaluation criteria. Lastly, the conclusion could be improved with a more explicit discussion of the results and the impact of the presented evaluations. In the current version, the insights from the evaluations do not seem to match the main experiments or the goals of the evaluation. (For more details, see the discussion of weaknesses.)

**Strengths And Weaknesses:**

# Strengths

- The proposed taxonomy appears to be novel and categorizes ECTS approaches in reasonable classes.
- The paper proposes a new simple baseline for ECTS.
- The visualizations presented in Figure 5 and Figure 8 intuitively illustrate the tradeoff between the two costs.

# Weaknesses

- A proper discussion of related work is missing. Therefore, it is difficult to assess the novelty of the proposed taxonomy and the evaluations.
- The paper is very opinionated and does not provide sufficient evidence for all its claims. For example, the claims about related work and their use of baselines, evaluation protocols, or datasets, or when the paper argues for the proposed evaluation protocol.
- The paper is not well-written.
  - The paper is repetitive. For example, the first two paragraphs of Section 3.1 repeat individual sentences and entire explanations detailed in the prior section. Another example are the cost functions, which are introduced multiple times throughout the paper.
  - The paper is inconsistent in its use of terminology and formalism. For example, some elements or notations are not properly defined, such as ECTS functions, and some minor aspects have dedicated equations, for example eq. (6) and (7), while the definition of a separable model does not.
  - The formatting is inconsistent. For example, the alignment of the figures.
  - The paper is often imprecise inviting additional questions. For example, is the length of each time series $T$ a global parameter of the dataset? If not, is there not also some temporal leakage for methods using this information?
- The evaluations are limited.
  - The evaluations compare 11 decision models of separable approaches. However, based on Table 1 there seem to be more methods than the ones evaluated here. Especially noticeable is the omission of more recent methods.
  - The datasets are selected based on their ability to be classified by one particular algorithm. Does this procedure not introduce selection bias? Additionally, it is not explained how the selection criteria are computed. The paper suggests the use of the proposed evaluation protocol for future evaluations. Does this apply to non-separable approaches? Has the second selection criteria been evaluated for the normalized datasets?
  - If only the decision functions are of interest for the evaluations, could we use synthetic classifiers instead?
  - The paper only presents aggregated results, which makes them of limited use for future evaluations and direct comparisons.
- There are no details on the proposed library.
- The survey of methods does not place each method properly in the proposed taxonomy and is generally inconsistent in the level of detail. Additionally, the survey appears incomplete, as other surveys consider more methods than the ones discussed here.

# Minor Comments

- It is a bit odd to call the second setting of the evaluation "anomaly detection", which is in itself a well-studied field that, especially on time series, significantly deviates from the setting considered here.
- Figures 1 and 2 are difficult to parse.
- Why do alap and asap not appear in the ranking evaluations?

---

> ### Author Response · Authors · 2025-09-26
>
> We thank the reviewer for his insightful comments and suggestions.  We answer in the order of the weaknesses mentioned:
>
> **W1** - *Position wrt. previous survey papers*: We have made more precise the position of this paper wrt. to previous survey papers underlining its originality at the end of the Introduction section under the title: *Position with respect to other literature surveys*.
>
> **W2** - *Opinionated  paper*: We have substantially changed the tone of the paper, in particular in Section 1 so that, we hope, it is no longer opinionated and is now a list of guidelines for evaluating ECTS systems.
>
> **W3** - *The paper is not well-written*:
>
> (*i*) We have largely pruned Sections 1 (Introduction) and 2 (*ECTS: concepts and taxonomy*) to eliminate the redundancies and remove unnecessary details, shortening the paper by 3 pages;
>
> (*ii*)  We had a pass on all equations to keep the important ones at the right place in the paper and all their definitions are in a single section "problem statement";
>
> (*iii*) We have been careful about the consistency of the formatting and we improved Figures 1 and 2 ;
>
> (*iv*) We gave more precision on the length of time series $T$ that is the same for all training examples for a given dataset (in the *Problem statement* section), thus averting any possibility of leakage.
>
> **W4** - *The evaluations are limited*:
>
> (*i*) The paper is intended to compare methods that are separable and representative of the properties highlighted in the proposed taxonomy. We choose papers according to both their representativeness and the fact that their code was available;
>
> (*ii*) Our objective was to compare trigger functions with everything else fixed, including the classifier, for which we chose one of the most popular, very scalable and accurate, TSC: MiniROCKET. Furthermore, we have checked ranking robustness to the choice of classifier (Appendix E.5) suggesting no manifest bias in the datasets selection. In addition, the selection criterion was clarified in Section 4.1.5 (paragraph *Dataset selection*). Notice that the proposed evaluation protocol can not be directly applied to end-to-end approaches and needs some adaptation, which is now mentioned as future work in the conclusion section. The selection criterion has not been evaluated for the z-normalized collection, as they were primarily used in order to reproduce past experiments.
>
> (*iii*) Synthetic classifiers could also be used for the evaluation and comparison of the ECTS methods. However, synthetic classifiers would present their own bias as well, and this is why, as mentioned before, we choose instead to use a popular classifier recognized as fast and scalable.
>
> (*iv*) The paper reports results that are statistically significant for the evaluation and comparison of the methods. Less aggregated measures could produce misleading results and conclusions. We will make available the detailed results of our main experiments with the open source library.
>
> **W5** - *Details on the library*:
> Appendix B, detailing the library formalism and logic has been added.
>
> **W6** - *The survey of methods does not place each method properly in the proposed taxonomy*:
> We have been attentive to take into account the observations about the structure of Sections 2 (*ECTS: concepts and taxonomy*) and 3 (*An organized state-of-the-art of separable methods*). The main concepts highlighted in Section 2 are now consistently used in Section 3 for the presentation of the methods. We clarified the scope of the paper, i.e. focusing on the separable ECTS approaches, which allows us to improve these two sections.
>
> **Minor comments**:
>
> **1** - Footnote 5 discusses the meaning of anomaly that is considered in the experiment section;
>
> **2** - Figures 1 and 2 have been adapted to be more easily understandable;
>
> **3** - The baselines ASAP and ALAP are not taken into account in the ranking figures, because it squashes the scale of these figures, hurting readability.
>
> **Other changes**:
> - The conclusion has been thoroughly rewritten to better reflect the content of the paper.
> - Even though the scope of the paper is about separable approaches, we have tested two end-to-end approaches and the related results are reported in Appendix A.2

---

> > ### Comment · Reviewer_zkBX · 2025-10-14
> >
> > The authors have made significant changes to the paper, which have improved the paper overall.
> >
> > Most of my concerns have been sufficiently addressed.
> >
> > The paper still seems to be missing a discussion of related work on benchmarking.

---

### Decision · Action_Editor_gCJ7 · 2025-10-28

**Recommendation:** Accept with minor revision

**Additional Comments:**

Reviewer zkBX remains concerned that

- the discussion of related work on benchmarking is still insufficient, and
- although the writing has improved, the paper remains too long and lacks adequate coverage of related studies.

I therefore request that the authors further address these issues in the revision.

**Audience:**

Yes

**Audience Explanation:**

Early classification of time series is an important problem, and the proposed taxonomy, library, and datasets will be valuable resources for both researchers and practitioners.

**Claims And Evidence:**

Yes

**Claims Explanation:**

This paper presents a principled taxonomy for early classification of time series and benchmarks state-of-the-art methods using a standardized evaluation protocol. The proposed taxonomy is both useful and novel compared to prior work.

The authors also provide an open-source library that implements most algorithms and includes curated datasets, significantly enhancing reproducibility and accessibility for future research.

---

> ### Author Response · Authors · 2025-11-17
>
> We thank the AC and all the reiewers for their time and valuable feedback.
>
> The following changes has been made in the camera-ready version:
> - Table 3 in Appendix A has been added as to account for benchmarking methodologies in related works.
> - Some redundancies has been removed, some expemrimental protocol details has been moved to Appendix to reduce paper length by nearly three more pages.
> - We've added a ``Position wrt other literature surveys'' at the end of Section 1.